# Backward-Compatible Prediction Updates: A Probabilistic Approach

**Frederik Träuble** [1][*][†]     **Julius von Kügelgen** [1,3][*]     **Matthäus Kleindessner** [2]

**Francesco Locatello** [2]     **Bernhard Schölkopf** [2]     **Peter Gehler** [2][†]

[1]Max Planck Institute for Intelligent Systems, Tübingen, Germany
[2]Amazon Tübingen, Germany
[3]Department of Engineering, University of Cambridge, United Kingdom

## Abstract

When machine learning systems meet real world applications, accuracy is only one of several requirements. In this paper, we assay a complementary perspective originating from the increasing availability of pre-trained and regularly improving state-of-the-art models. While new improved models develop at a fast pace, downstream tasks vary more slowly or stay constant. Assume that we have a large unlabelled data set for which we want to maintain accurate predictions. Whenever a new and presumably better ML models becomes available, we encounter two problems: (i) given a limited budget, which data points should be re-evaluated using the new model?; and (ii) if the new predictions differ from the current ones, should we update? Problem (i) is about compute cost, which matters for very large data sets and models. Problem (ii) is about maintaining consistency of the predictions, which can be highly relevant for downstream applications; our demand is to avoid negative flips, i.e., changing correct to incorrect predictions. In this paper, we formalize the Prediction Update Problem and present an efficient probabilistic approach as answer to the above questions. In extensive experiments on standard classification benchmark data sets, we show that our method outperforms alternative strategies along key metrics for backward-compatible prediction updates.

## 1 Introduction

The machine learning (ML) community develops new models at a fast pace: for example, just in the past year, the state-of-the-art on ImageNet has changed at least five times [9, 10, 31, 50, 51]. As reproducibility has increasingly been scrutinized [33, 34, 42], it is now common practice to release pre-trained models upon publication. In this work we take the perspective of an owner of an unlabelled data set who is interested in keeping the best possible predictions at all times. When a new pre-trained model is released, we face what we refer to as the *Prediction Update Problem*: (i) decide which points in the data set to re-evaluate with the new model, and (ii) integrate the new, possibly contradicting, predictions. For this task, we postulate the following three desiderata:

1. The prediction updates should improve overall accuracy.
2. The prediction updates should avoid introducing new errors.
3. The prediction updates should be as cheap as possible since the target data set could be huge.

---

[*]Work done while FT and JvK were interning at Amazon.
[†]Correspondence to: `frederik.traeuble@tuebingen.mpg.de` and `pgehler@amazon.com`

35th Conference on Neural Information Processing Systems (NeurIPS 2021).

We consider the setting in which the target data set for which we wish to maintain predictions is fully unlabelled (i.e., the ground-truth labels are unknown) and may come from a different distribution than the one on which models have been pre-trained, but with overlap in the label space. This is a transductive or semi-supervised problem, but, due to computational constraints, we avoid any model fitting or fine-tuning and rely solely on the predictions of the pre-trained models that are released over time. Typically, these models exhibit increased performance on their labelled training domain (e.g., the ImageNet validation or test set) as evidence for being good candidates for re-evaluation.

Clearly, one goal of updating the predictions stored for the target data set is to improve overall performance, e.g., top-k accuracy for classification. At the same time, the stored predictions may form an intermediate step in a larger ML pipeline or are accessible to users. This is the reason for our second desideratum: we would like to be *backward-compatible*, i.e., new predictions should not flip previously correct predictions (*negative flips*). Finally, we aim to reduce computational cost during inference and to avoid evaluating the entire data set which may be prohibitive in practice and unnecessary if we are already somewhat certain about a prediction.

In this paper, we motivate and formalize the *Prediction Update Problem* and describe its relation to various relevant research areas like ensemble learning, domain adaption, active learning, and others. We propose a probabilistic approach that maintains a posterior distribution over the unknown true labels by combining all previous model re-evaluations. Based on these uncertainty estimates, we devise an efficient *selection strategy* which only chooses those examples with highest posterior label entropy for re-evaluation in order to reduce computational cost. Furthermore, we consider different prediction-update strategies to decide whether to change the stored predictions, taking asymmetric costs for negative and positive flips into account. Using the task of image classification as a case study, we perform extensive experiments on common benchmarks (ImageNet, CIFAR10, and ObjectNet) and demonstrate that our approach achieves competitive accuracy and introduces much fewer negative flips across a range of computational budgets, thus showing that our three desiderata are not necessarily at odds.

**Contributions** We highlight the following contributions:

- We introduce the *Prediction Update Problem* which addresses some common, but previously unaddressed challenges faced in real world ML systems (§ 2).
- We propose a probabilistic, model-agnostic approach for the *Prediction Update Problem*, based on Bayesian belief estimates of the true label combined with an efficient selection and different prediction-update strategies (§ 3).
- We contextualise this understudied problem setting as well as our method with related work (§ 4 & § 5) and discuss several extensions and limitations (§ 4).
- We demonstrate that our approach successfully outperforms alternative approaches and accomplishes all our desiderata in experiments across multiple common benchmark datasets (CIFAR-10, ImageNet, and ObjectNet) and practically relevant scenarios (§ 6).[3]

## 1.1 Backward-Compatible Prediction Systems

In real world ML applications, empirical performance is only one of several requirements. When humans interact with automatic predictions, they will start to build mental models of how these models operate and whether and when their predictions can be trusted. This is described as Human-AI teams by [1] who argue to "make the human factor a first-class consideration of AI updates".

An example from [1] is autopilot functionality in cars for which drivers will build expectations in which driving situations the autopilot is safe to engage. It is important not to violate these assumptions when updating the models over the air. AI assisted medical decision processes are another example of a high stake application where medical professionals need to understand when systems can be trusted.

Consider the example of automatically tagging images in a user's photo collection. Those tags are used for example in photo search. As models progress, the overall accuracy on all uploaded images may increase, but for any single user the experience can deteriorate if previously correct searches now show wrong results. Even worse, if errors fluctuate over the user's photo collection as the result of prediction-updates, the user's trust will be eroded. This "cost" is asymmetric and the negative experience may outweigh the benefit of better predictions on other images.

---

[3]All software and assets we use are open source and under MIT, Apache or Creative Commons Licenses.

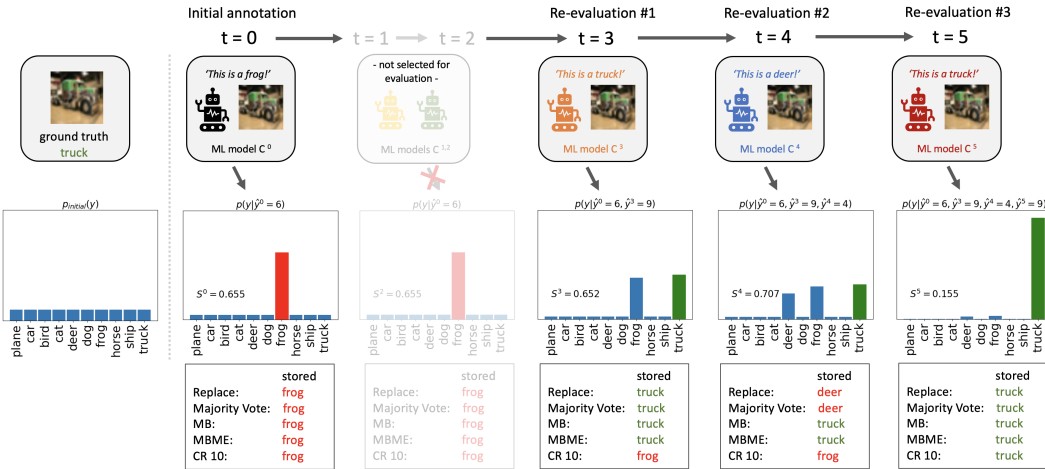

Figure 1: **Overview of our proposed Bayesian approach to the Prediction Update Problem.** Starting from a uniform prior, we maintain a posterior distribution $p(y = k|\hat{y}^{0:t})$ (*middle*) over the unknown true label $y$ of an unlabelled sample which takes the predictions $\hat{y}^{0:t}$ from new ML models $C^t$ (*top*) arriving over time $t = 0, ..., T$ into account. Given a limited compute budget $B^t$, we re-evaluate those samples with highest posterior label entropy $S^t$ at each time step, e.g., the example shown is first selected in time step $t = 3$ after the initial annotation at $t = 0$. We then consider different strategies for deciding whether to update the stored prediction (*bottom*) based on our changed beliefs. Note that the non-probabilistic baselines "Replace" (always update to last prediction) and "Majority Vote" (resolve ties by using the last prediction) incorrectly update the stored prediction from "truck" to "deer" in step $t = 4$. Our strategies (MB, MBME, CR-10) which rely on the estimated label posterior, on the other hand, avoid such a *negative flip*, which is one of our key goals.

In contrast to carefully curated and labelled ML benchmarks, many real-world data sets are magnitudes larger (up to billions of samples) and entirely unlabelled. Having no feedback which predictions are correct is a common scenario: consider any type of private data such as health data, photo collections, or personal information. Because the data is private, we can neither train on it, nor collect feedback, nor observe the effect of predictions. On the other hand, such data is valuable to an individual: she has an interest to keep it up to date with the best possible predictions. Since it is of little consolation to her if an update of the model improves predictions on average but on her data it gets worse, the update costs are asymmetric. Service providers often rely on models pre-trained on a different data set, and the desire to be backward-compatible arises naturally in this setting [1, 39, 43, 53].

This is an understudied problem where progress will have large impact. ML systems are becoming pervasive, and their accuracy will continue to increase. Being able to seamlessly transfer them to existing data will be crucial for real-world ML systems.

## 2    The Prediction Updates Problem Setting

**Target Data Set**   We are given a large, unlabelled target data set $\mathcal{D}^{\text{targ}} = \{\mathbf{x}_n\}_{n=1}^N$ comprising $N$ independent and identically distributed (i.i.d.) observations $\mathbf{x}_n \in \mathcal{X} \subseteq \mathbb{R}^d$ drawn from a target distribution $\mathbb{P}_{\mathbf{X}}^{\text{targ}}$. The ground-truth labels $y_n \in \mathcal{Y} = \{1, ..., K\}$ distributed according to $\mathbb{P}_{Y|\mathbf{X}}^{\text{targ}}$ are *not observed*. Note that we are particularly interested in a scenario where $N$ may be extremely large.

**Models**   Over time $t = 0, 1, ..., T$ we successively gain access to classifiers $C^0, C^1, \ldots, C^T : \mathcal{X} \to \mathcal{Y}$ which have been trained on a labelled data set $\mathcal{D}^{\text{src}}$ from a potentially different source distribution $\mathbb{P}_{\mathbf{X},Y}^{\text{src}}$ over $\mathcal{X} \times \mathcal{Y}$. For simplicity, we assume that the observation space $\mathcal{X}$ and label space $\mathcal{Y}$ are shared. We consider both the standard scenario where the models $\{C^t\}_{t=0}^T$ are trained on a labelled set from the same domain ($\mathbb{P}^{\text{src}} = \mathbb{P}^{\text{targ}}$); and the transfer scenario where we deploy a model trained on a labelled ML benchmark to a different data set ($\mathbb{P}^{\text{src}} \neq \mathbb{P}^{\text{targ}}$). We assume that $\{C^t\}_{t=0}^T$ are improving in performance on the training data set. Therefore, denoting by $A_t$ the estimated accuracy of $C^t$ on $\mathbb{P}_{\mathbf{X},Y}^{\text{src}}$, we have $A^t \le A^{t+1} \forall t$. As motivating example, consider an object recognition task in the wild and let $C_t$ be the winning entry of the ImageNet competition in year $t$.

**Labelling** To relate the source and target distributions and to justify applying $\{C^t\}_{t=0}^T$ to our target data set $\mathcal{D}^{\text{targ}}$, we make the commonly used covariate shift assumption $\mathbb{P}_{Y|\mathbf{X}}^{\text{targ}} = \mathbb{P}_{Y|\mathbf{X}}^{\text{src}}$, i.e. the conditional label distribution is shared across source and target distributions [40, 45].[4] We denote the *predicted* label by $C^t$ for $\mathbf{x}_n$ by $\hat{y}_n^t = C^t(\mathbf{x}_n)$ and the *stored* prediction for $\mathbf{x}_n$ after time step $t$ by $l_n^t$. The target data set is then initially fully labelled by $C^0$, i.e., $l_n^0 := \hat{y}_n^0$.

**Objective** As new classifiers $\{C^t\}_{t\geq 1}$ become available, our main objective is to maintain the best estimates $\{l_n^t\}_{n=1}^N$ on our target data set at all times and improve overall accuracy, while, at the same time, maintaining backward compatibility by minimising the number of *negative flips*, i.e., the number of previously correctly stored predictions that are incorrectly changed. The key challenge is that no ground truth labels for our target data set are available, so that we have no feedback on which predictions are correct and which are wrong. For each test sample $\mathbf{x}_n$ and each time step $t \geq 1$, we thus need to decide whether or not to update the previously stored prediction $l_n^{t-1}$ based on the current and previous model predictions $\hat{y}_n^t$ and $\hat{y}_n^{0:t-1}$, respectively.

**Limited Evaluation Budget** Re-evaluating all samples (so-called *backfilling*) can be very costly and requires significant resources. Since we consider $N$ to be very large, we also consider a limited budget of at most $B^t \leq N$ sample re-evaluations for step $t$. We thus additionally need to decide how to allocate this budget and select a subset of samples to be re-evaluated by $C^t$ at every step.

## 3 Our Method

Having specified the setting, we next describe our proposed method for the Prediction Update Problem. We start by providing a Bayesian approach for maintaining and updating our beliefs about the unknown true labels as new predictions become available (§ 3.1), followed by describing strategies for selecting candidate samples for re-evaluation (§ 3.2) and for updating the stored predictions based on our changed beliefs (§ 3.3). Our framework is summarised in Figure 1.

### 3.1 Bayesian Approach

Since the true labels $\{y_n\}_{n=1}^N$ are unknown to us, we treat them as random quantities over which we maintain uncertainty estimates. We then perform Bayesian reasoning to update our beliefs as new evidence in the form of predictions $\hat{y}_n^t$ from newly-available classifiers $C^t$ arrives over time $t = 1, ..., T$. In standard Bayesian notation, the true labels $y_n$ thus take the role of unknown parameters $\theta$ and the predictions $\hat{y}_n^t$ of data $x$. Since $\mathcal{D}^{\text{targ}}$ is sampled i.i.d., we reason about each label $y_n$ independently of the others, i.e., the following is the same for all $n$.

**Prior** Lacking label information on the target data set, we choose a uniform prior over $\mathcal{Y}$ for all $y_n$, i.e., $p(y_n = k) = 1/K$, $\forall k \in \mathcal{Y}$. If (estimates of) the class probabilities on $\mathcal{D}^{\text{targ}}$ are available, we may instead use these as a more informative prior.

**Likelihood** Next, we need to specify a likelihood function $p(\hat{y}_n^{0:T}|y_n = k)$ for the observed model predictions $\hat{y}_n^{0:T}$ given a value $k$ of the true label $y_n$. We make the following simplifying assumption.

**Assumption 1** (Conditionally independent classifiers). *The different classifiers' predictions $\hat{y}_n^{0:T}$ are conditionally independent given the true label $y_n$, i.e., the likelihood factorises as*

$$p(\hat{y}_n^{0:T}|y_n = k) = \prod_{t=0}^T p(\hat{y}_n^t|y_n = k). \tag{1}$$

In a standard Bayesian setting, this corresponds to the assumption of conditionally independent observations given the parameters; we refer to § 4 for further discussion. The main advantage of Assumption 1 is that the factors $p(\hat{y}_n^t|y_n = k)$ on the RHS of (1) have a natural interpretation: these are the (normalised) confusion matrices $\pi^t$ of the classifiers $C^t$, i.e., we denote by

$$\pi^t(i, k) := p(\hat{y}^t = i|y = k),$$

the probability that $C^t$ predicts class $i$ given that the true label is $k$, which is the same for all $n$; see below and § 4 for more details on how we estimate $\pi^t$ in practice.

---

[4] In the context of image classification, this means that images of the same object under different environmental conditions (i.e., different $\mathbb{P}_{\mathbf{X}}$) will always share the same label (i.e., same $\mathbb{P}_{Y|\mathbf{X}}$). Note that such covariate shift may also lead to changes in $\mathbb{P}_Y$ (label/target shift) and/or $\mathbb{P}_{\mathbf{X}|Y}$ (conditional shift) [44, 55].

**Posterior** At every time step $t \geq 0$, we can then compute our posterior belief about the true label $y_n$ given model predictions $\hat{y}_n^{0:t}$ according to Bayes rule,

$$p(y_n = k | \hat{y}_n^{0:t}) = \frac{\pi^t(\hat{y}_n^t, k) p(y_n = k | \hat{y}_n^{0:t-1})}{\sum_{i \in \mathcal{Y}} \pi^t(\hat{y}_n^t, i) p(y_n = i | \hat{y}_n^{0:t-1})} \tag{2}$$

where we have used Assumption 1 to write $p(\hat{y}_n^t | y_n = k, \hat{y}_n^{0:t-1}) = p(\hat{y}_n^t | y_n = k) = \pi^t(\hat{y}_n^t, k)$. The posterior at step $t-1$ acts as prior for step $t$, so we do not have to store all previous predictions.

**Estimating Confusion Matrices** In practice, $\pi^t$ are generally not known and we instead use their (maximum likelihood) estimates $\hat{\pi}^t$ from the source distribution. If the number of classes $K$ is large compared to the amount of labelled source data,[5] we only estimate the diagonal elements $\hat{\pi}_{kk}^t$ (i.e., the class-specific accuracies) and set the $K(K-1)$ off-diagonal elements to be constant,

$$\hat{\pi}^t(i, k) = \frac{1 - \hat{\pi}^t(k, k)}{K - 1} \qquad \forall i \neq k,$$

so that $\sum_{i=1}^K \hat{\pi}^t(i, k) = 1 \; \forall k \in \mathcal{Y}$. We refer to § 4 for further discussion on the estimation of $\pi^t$.

### 3.2 Selecting Candidates for Re-evaluation

Given the label posteriors computed according to (2), we compute the Shannon entropies [38]

$$S_n^t = - \sum_{k \in \mathcal{Y}} p(y_n = k | \hat{y}_n^{0:t}) \log p(y_n = k | \hat{y}_n^{0:t}),$$

which provide a simple measure of uncertainty in the true label $y_n$ after step $t$. We then select and re-evaluate the $B^t$ samples with highest posterior label entropy $S_n^t$ to update our beliefs.

### 3.3 Prediction-Update Strategies

Finally, we need a strategy for deciding whether and how to update the previously stored prediction $l_n^{t-1}$ based on our new beliefs. We consider three such prediction-update strategies.

**MaxBelief (MB)** The simplest approach is to always update based on the maximum a posteriori belief, i.e., $l_n^t := \hat{l}_n^t = \text{argmax}_{k \in \mathcal{Y}} \, p(y_n = k | \hat{y}_n^{0:t})$. We refer to this strategy as MaxBelief (MB).

**MaxBeliefMinEntropy (MBME)** A slightly more sophisticated approach is to also take the change in posterior entropy into account and only update when it has decreased:

$$l_n^t := \begin{cases} \hat{l}_n^t & \text{if} \quad S_n^t < S_n^{t-1}, \\ l_n^{t-1} & \text{otherwise.} \end{cases}$$

We refer to this strategy as MaxBeliefMinEntropy (MBME).

**CostRatio (CR)** So far, we have not taken the assumed larger penalty for negative flips into account. We therefore now develop a third approach based on asymmetric flip costs. We denote the cost of a negative flip (NF) by $c^{\text{NF}} > 0$ and that of a positive flip (PF) by $c^{\text{PF}} < 0$.

We need to decide whether to update the previously stored prediction $l_n^{t-1}$ based on our updated beliefs $p(y_n = k | \hat{y}_n^{0:t})$. Denote the MAP label estimate after step $t$ by $\hat{l}_n^t = \text{argmax}_{k \in \mathcal{Y}} \, p(y_n = k | \hat{y}_n^{0:t})$. If $\hat{l}_n^t = l_n^{t-1}$ there is no reason to change the stored prediction. Suppose that $\hat{l}_n^t \neq l_n^{t-1}$. We then need to reason about the (estimated) positive and negative flip probabilities when changing the stored prediction from $l_n^{t-1}$ to $\hat{l}_n^t$. A positive flip (PF) occurs if $\hat{l}_n^t$ is the correct label (and hence $l_n^{t-1}$ is not), and, vice versa, a negative flip occurs if $l_n^{t-1}$ is correct (and hence $\hat{l}_n^t$ is not):

$$\hat{p}_n^{\text{PF}}(l_n^{t-1} \rightarrow \hat{l}_n^t) = p(y_n = \hat{l}_n^t | \hat{y}_n^{0:t}), \qquad \hat{p}_n^{\text{NF}}(l_n^{t-1} \rightarrow \hat{l}_n^t) = p(y_n = l_n^{t-1} | \hat{y}_n^{0:t}).$$

If neither $l_n^{t-1}$ nor $\hat{l}_n^t$ are the correct label, the flip is inconsequential which we assume incurs zero cost. The estimated cost of changing the stored prediction from $l_n^{t-1}$ to $\hat{l}_n^t$ is thus:

$$\hat{c}(l_n^{t-1} \rightarrow \hat{l}_n^t) = c^{\text{NF}} \hat{p}_n^{\text{NF}}(l_n^{t-1} \rightarrow \hat{l}_n^t) + c^{\text{PF}} \hat{p}_n^{\text{PF}}(l_n^{t-1} \rightarrow \hat{l}_n^t).$$

---

[5]For example, on ImageNet we have $K = 1000$ which would require estimating 1 million parameters.

We only want to change the prediction if $\hat{c}(l_n^{t-1} \to \hat{l}_n^t) < 0$, i.e.,

$$\frac{\hat{p}_n^{\mathrm{PF}}(l_n^{t-1} \to \hat{l}_n^t)}{\hat{p}_n^{\mathrm{NF}}(l_n^{t-1} \to \hat{l}_n^t)} = \frac{p(y_n = \hat{l}_n^t | \hat{y}_n^{0:t})}{p(y_n = l_n^{t-1} | \hat{y}_n^{0:t})} > -\frac{c^{\mathrm{NF}}}{c^{\mathrm{PF}}} \tag{3}$$

leading to the following update rule:

$$l_n^t := \begin{cases} \hat{l}_n^t, & \text{if} \quad \hat{l}_n^t = l_n^{t-1}, \\ \hat{l}_n^t, & \text{if} \quad \hat{l}_n^t \neq l_n^{t-1} \wedge \hat{c}(l_n^{t-1} \to \hat{l}_n^t) < 0, \\ l_n^{t-1} & \text{otherwise.} \end{cases}$$

Note that (3) has an intuitive interpretation: we only want to update the currently stored prediction (thus potentially risking a negative flip) if our belief in a different label is larger than that in the current one by a factor exceeding $|c^{\mathrm{NF}}/c^{\mathrm{PF}}|$. We therefore refer to this strategy as CostRatio (CR).

## 4  Discussion: Extensions and Limitations

We discuss current limitations of our method and propose extensions to address them in future work.

**Soft vs. Hard Labels**  Our approach presented in § 3 assumes deterministic classifiers which output hard labels, i.e., only the most likely class. This allows for maximum flexibility and a wide range of classifier models that can be used in conjunction with this method. However, our Bayesian framework can easily be adapted to also allow for probabilistic classifiers which output soft labels, i.e., vectors of class probabilities. We included some additional exploratory experiments utilizing softmax probabilities in Appendix A.11. Since deep neural networks are known to have unreliable uncertainty estimates [14, 25, 26, 47], we deliberately choose to work with hard labels. If, however, well-calibrated probabilistic classifiers are available (and can be scaled to huge data sets), taking this additional information into account will likely lead to more accurate posterior estimates and thus better performance.

**Assumption of Conditionally-Independent Classifiers**  Since the models $\{C^t\}$ are typically trained and developed on the same data and may even build on insights from prior architectures, our assumption of conditionally independent predictions on $\mathcal{D}^{\mathrm{targ}}$ does likely not hold exactly in practice. It should therefore rather be understood as an approximation that enables tractable posterior inference. Our experiments (§ 6) suggest that it is a useful approximation that yields competitive performance. Properly incorporating estimated model correlations may yield further improvements. We refer the interested reader to Appendix A.1 for a more detailed discussion.

**Confusion Matrix Estimates**  Unless labelled data from $\mathbb{P}^{\mathrm{targ}}$ is available, the confusion matrices $\{\pi^t\}$ need to be estimated from $\mathbb{P}^{\mathrm{src}}$. This is only an approximation because they may change as a result of $\mathbb{P}_{\mathbf{X}}^{\mathrm{src}} \neq \mathbb{P}_{\mathbf{X}}^{\mathrm{targ}}$, and taking such shifts into account could yield more accurate posterior estimates. For this, one may use ideas from the field of *unsupervised domain adaptation* [12, 28, 45]. One could use an importance-weighting approach [40] to give more weight to points which are representative of $\mathbb{P}_{\mathbf{X}}^{\mathrm{targ}}$ when estimating $\pi^t$ from $\mathbb{P}_{\mathbf{X},Y}^{\mathrm{src}}$. As an example, in further experiments in Appendix A.4 we studied estimating the off-diagonal elements using Laplace smoothing [13, 35],

**Other Selection Strategies**  Consider an ambiguous image that could be either a zucchini or a cucumber [4]. Such a sample would have large label entropy and could thus potentially be selected for re-evaluation again and again. To overcome this hurdle, one could decompose label uncertainty into epistemic (reducible) and aleatoric (irreducible) uncertainty [8, 19] and only re-evaluate samples with high aleatoric uncertainty, i.e., those with high expected information gain [24]. Such considerations also play a role in the field of *active learning* [37, 54]

**Growing Data Set Size**  Our method is not constrained to fixed data set sizes and can accommodate for the addition of new data. New samples can be added at any time using an uniform prior over labels. Given their high initial entropy, they would then be naturally selected for (re-)evaluation first.

**Adaptive Budgets**  Currently, we consider a fixed local budget of $B^t$ re-evaluations at every time step. A possible extension would be to allow for a global budget of $B^{\mathrm{total}}$ evaluations spread over all time-steps, i.e., to devise a strategy for deciding whether to (a) keep re-evaluating or (b) save budget for the next better model, potentially using techniques from reinforcement learning [46].

**On the Cost of "Neutral" Flips**  For simplicity, we have assumed that "neutral" flips (i.e., changing a label estimate from an incorrect to a different incorrect one) bear no cost. However, as motivated

in § 1, it is well conceivable that even such neutral flips have a cost due to the potential to disrupt downstream robustness. If this is the case, it can easily be incorporated into our CR update strategy.

## 5 Related Work

Besides the aforementioned connections, our problem setting bears resemblance to several other areas of ML. In the following, we discuss the main differences and commonalities.

**Backward Compatibility** The term was first introduced by Bansal et al. [1] in the context of humans making decisions based on an AI's prediction (e.g., medical expert systems or driver supervision in semi-autonomous vehicles). They contextualise that even though an AI's predictive performance might increase overall, *incompatible* predictions in updated models severely hurt overall performance and trust, and propose to penalize negative flips w.r.t an older model when training a newer model. Yan et al. [53] show that with standard training, there can be a significant number of negative flips, even if the two models only differ in their random initializations. They then reduce the number of negative flips by giving more weight to training points that are correctly classified by the reference model, which they call 'positive-congruent training'. Previous work on backward-compatible learning is concerned with training a *new* model. Here, we focus on updating the stored predictions rather than updating the stored models. This makes our approach more generally applicable and complements the use-cases of backward-compatible learning. Backward compatibility was further studied empirically by Srivastava et al. [43] who emphasize that this also causes problems for large multi-component AI systems. They propose two key metrics to characterize backward compatibility: (i) Backward Trust Compatibility (BTC), first mentioned in [1], measuring the fraction of predictions that are still predicted correctly after a model update; and (ii) Backward Error Compatibility (BEC), which corresponds to the probability that an incorrect prediction after an update is not new.

**Ensemble Learning** Ensemble methods aim to combine several ML models into a single model with higher performance than each of the individual models. Common techniques are boosting [11], bagging [6], or Bayesian model averaging [16]. Our approach falls into the latter category. We compute the posterior probability (2) in the same way as the well-known Naive Bayes combiner [23]. The classifier corresponding to our MB strategy goes back to at least Nitzan and Paroush [27] and has been thoroughly analyzed [3]. There are also Bayesian techniques that avoid Assumption 1, but these either make some parametric assumptions [20] or assume a very special form of dependence [5].

## 6 Experiments

We now evaluate our Bayesian approach to the Prediction Update Problem against different baselines using the task of image classification as a case study.

### 6.1 Experimental Setup

**Data Sets** We use the three widely accepted benchmark data sets ImageNet1K [7] (1K classes, 50k validation set), ObjectNet [2] (313 classes, 50k validation set) and CIFAR-10 [21] (10 classes, 10k validation set). To imitate our assumed setting of deploying pre-trained models to an unlabelled target data set, we only use the corresponding validation sets as $\mathcal{D}^{\text{targ}}$. The ground truth labels are only used post-hoc to compute performance metrics and are not seen during the $T$ update steps. Of the 313 classes in ObjectNet, 113 are shared with ImageNet, corresponding to a subset of 18,547 images. ObjectNet images exhibit more realistic variations than those in ImageNet. It only has a test set and thus constitutes a challenging transfer scenario for object recognition models. We deploy ImageNet-pretrained models both on ImageNet and on the above subset of ObjectNet, thus simulating the cases that the source and target distributions are the same or different, respectively. For the former, we split the ImageNet validation set in half and use one half to estimate $\pi^t$ and the other as $\mathcal{D}^{\text{targ}}$. For the latter, we estimate $\pi^t$ from the full ImageNet validation set and evaluate on ObjectNet. We thus assume the covariate shift case, where the conditional label distribution is shared across source and target distributions.

**Models & Architectures** To emulate the setting of sequentially improving classifiers arriving over time, we use the following 17 models and architectures with many of them setting a new "state-of-the-art" on ImageNet at the time they were first introduced: AlexNet [22]; VGG-11, 13, 16, and 19 [41]; ResNet-18, 34, 50, 101, and 152 [15]; SqueezeNet [18]; GoogLeNet [48];

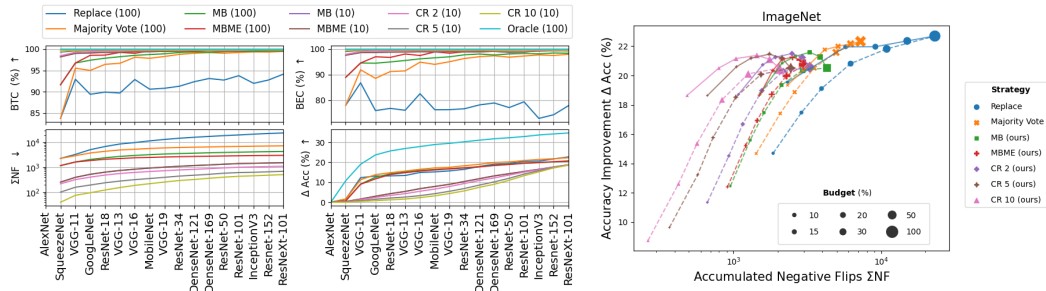

Figure 2: **Left:** Temporal evolution for ImageNet $\rightarrow$ ImageNet over $T = 16$ prediction-update steps for a subset of strategies and budgets. **Right:** Comparison of prediction-update strategies across different budgets after $T = 16$ prediction-update steps. Dashed lines correspond to the ablation using a random selection strategy.

InceptionV3 [49]; MobileNetV2 [36]; DenseNet-121 and 169 [17], and ResNeXt-101 32x8d [52]. For ease of reproducibility, we use pre-trained models from the torchvision model zoo [29] and [32].

**Performance Metrics**   Recall that our goal is to: (i) improve overall accuracy, (ii) avoid negative flips, and (iii) use as few re-evaluations as possible. To assess these different aspects, we report the following metrics: (i) *final accuracy* of the stored predictions (**Acc**) and *accuracy improvement* over the initial accuracy of $C^0$ ($\Delta$**Acc**); (ii) the *cumulative number of negative flips* from time $t = 0$ to $T$ ($\Sigma$ **NF**), the *average negative flip rate* experienced per iteration, i.e., $\frac{\Sigma\text{ NF}}{N \cdot T}$ (**NFR**), and the ratio of accumulated positive to negative flips (**PF/ NF**); (iii) the evaluation budget available to each strategy as percentage of the data set size, i.e., a budget of 10 means that 10% of all samples can be re-evaluated at each time step: $B^t = 0.1N, \forall t$; finally, we measure the connective backward compatibility between (i) and (ii) via Backward Trust Compatibility (**BTC**) and Backward Error Compatibility (**BEC**) [43]. We refer to Appendix A.2 for a formal definition of these scores.

**Baselines and Oracle**   We compare our method against two baselines: (i) **Replace** always updates the stored prediction with that predicted by the most recent classifier (a.k.a. *backfilling*); (ii) **Majority Vote** takes into account previous model predictions and updates the stored prediction according to the majority prediction. In case of a tie, the prediction of the most recent classifier is chosen. For reference, we also compare our method against an **Oracle**, which performs a prediction update if and only if this would lead to a positive flip; it thus incurs zero negative flips by definition (knowing the ground truth label). We emphasize that, in practice, we do not have that information in our setting.

**Selection- and Prediction-Update Strategies**   For all methods, we select $B^t \leq N$ samples using the posterior label entropy selection strategy from § 3.2, but also compare with randomly selecting samples for re-evaluation. For all experiments involving the random selection strategies we run the same experiment for five different random seeds each and report the average. We use the prediction-update strategies MB, MBME and CR from § 3.3 and consider cost ratios of $|c^{\text{NF}}/c^{\text{PF}}| \in \{2, 5, 10\}$ for the latter (e.g., CR 2).

## 6.2   Results for ImageNet $\rightarrow$ ImageNet

In Fig. 2 (left), we show the temporal evolution of backwards compatibility scores, negative flips and accuracy gains for prediction-updates on the ImageNet validation set for a subset of strategies and budgets. A complete account of final performances with additional metrics is shown in Tab. 1 (left).

For the evolution of $\Delta$**Acc** in Fig. 2 (left), we observe that, unsurprisingly, strategies with 100% budget experience a more rapid gain in accuracy than those with 10%. Among the budget-constrained strategies, the CR strategy with large cost ratio shows the slowest increase, which makes sense as it requires a substantial change in posterior belief for updating a stored prediction and is thus more conservative. Interestingly, however, the *final* accuracies only differ marginally across both strategies and budgets which is also apparent from the minor differences in the $\Delta$**Acc** column of Tab. 1. For the evolution of $\Sigma$ **NF** in Fig. 2 (right), we observe a clear separation of strategies with a natural ordering from least conservative (Replace) to most conservative (CR 10). These relative differences stay mostly constant over time as NFs appear to accumulate approximately linearly (note the log-scale). We find roughly an order of magnitude difference in $\Sigma$ **NF** between the best non-probabilistic baseline (Majority Vote) and the best Bayesian method (CR 10). Especially for small budgets of up to 30%,

Table 1: Results for ImageNet → ImageNet (left) and ImageNet → ObjectNet (right): all metrics refer to final performance for the improving model sequence from Fig. 2 and Fig. 3 respectively. The character **E** or **R** in front of the strategy indicates that selection for re-evaluation is based on the entropy criterion or sampled randomly.

**ImageNet → ImageNet**

| | Strategy | Avg. BTC ↑ | Avg. BEC ↑ | Acc (%) ↑ | ΔAcc (%) ↑ | Σ NF ↓ | NFR (%) ↓ | PF / NF ↑ |
|---|---|---|---|---|---|---|---|---|
| | Oracle | 100 | 100 | 91.2 | 34.7 | 0 | 0 | - |
| Budget = 100% | Replace | 91.37 | 77.71 | **79.2** | **22.7** | 24214 | 6.05 | 1.2 |
| | Majority Vote | 97.18 | 93.95 | 78.9 | 22.3 | 7352 | 1.84 | 1.8 |
| | MB | 98.32 | 96.45 | 77.1 | 20.5 | 4378 | 1.09 | 2.2 |
| | MBME | 98.78 | 97.69 | 77.3 | 20.7 | 3057 | 0.76 | 2.7 |
| | CR 2 | 98.72 | 97.19 | 77.1 | 20.6 | 3368 | 0.84 | 2.5 |
| | CR 5 | 99.06 | 97.82 | 77.1 | 20.5 | 2520 | 0.63 | 3 |
| | CR 10 | **99.22** | **98.15** | 77 | 20.5 | **2112** | **0.53** | **3.4** |
| Budget = 30% | R:Replace | 97.56 | 94.53 | 77.4 | 20.8 | 6546.4 | 1.64 | 1.8 |
| | R:Majority Vote | 98.6 | 97.18 | 77.1 | 20.5 | 3616.4 | 0.9 | 2.4 |
| | E:Replace | 96.53 | 91.01 | **78.5** | **22** | 9708 | 2.43 | 1.6 |
| | E:Majority Vote | 98.03 | 95.63 | **78.5** | **22** | 5232 | 1.31 | 2.1 |
| | E:MB | 98.71 | 97.25 | 78.1 | 21.6 | 3275 | 0.84 | 2.6 |
| | E:MBME | 98.98 | 98.04 | 77.8 | 21.2 | 2577 | 0.64 | 3.1 |
| | E:CR 2 | 99.02 | 97.86 | 78 | 21.5 | 2578 | 0.64 | 3.1 |
| | E:CR 5 | 99.32 | 98.43 | 78 | 21.5 | 1831 | 0.46 | 3.9 |
| | E:CR 10 | **99.44** | **98.67** | 77.9 | 21.4 | **1517** | **0.38** | **4.5** |
| Budget = 10% | R:Replace | 99.22 | 98.63 | 71.3 | 14.7 | 1958.4 | 0.49 | 2.9 |
| | R:Majority Vote | 99.4 | 98.98 | 71.2 | 14.7 | 1481.4 | 0.37 | 3.5 |
| | E:Replace | 99.04 | 98.12 | **76.1** | **19.5** | 2468 | 0.62 | 3 |
| | E:Majority Vote | 99.06 | 98.18 | 75.9 | 19.3 | 2417 | 0.6 | 3 |
| | E:MB | 99.38 | 98.89 | 75.3 | 18.8 | 1557 | 0.39 | 4 |
| | E:MBME | 99.38 | 98.92 | 75.2 | 18.7 | 1533 | 0.38 | 4 |
| | E:CR 2 | 99.55 | 99.22 | 75.3 | 18.7 | 1118 | 0.28 | 5.2 |
| | E:CR 5 | 99.72 | 99.51 | 75.2 | 18.6 | 700 | 0.18 | 7.7 |
| | E:CR 10 | **99.79** | **99.64** | 75.2 | 18.6 | **515** | **0.13** | **10.1** |

**ImageNet → ObjectNet**

| | Strategy | Avg. BTC ↑ | Avg. BEC ↑ | Acc (%) ↑ | ΔAcc (%) ↑ | Σ NF ↓ | NFR (%) ↓ | PF / NF ↑ |
|---|---|---|---|---|---|---|---|---|
| | Oracle | 100 | 100 | 50.5 | 42.6 | 0 | 0 | - |
| Budget = 100% | Replace | 72.65 | 92.61 | **31.9** | **24** | 16669 | 5.62 | 1.3 |
| | Majority Vote | 89.99 | 98.02 | 29.6 | 21.6 | 4690 | 1.58 | 1.9 |
| | MB | 94.46 | 98.96 | 29.1 | 21.2 | 2477 | 0.83 | 2.6 |
| | MBME | 95.86 | 99.34 | 28.6 | 20.6 | 1599 | 0.54 | 3.4 |
| | CR 2 | 95.92 | 99.21 | 29 | 21 | 1876 | 0.63 | 3.1 |
| | CR 5 | 97.18 | 99.41 | 28.8 | 20.8 | 1372 | 0.46 | 3.8 |
| | CR 10 | **97.82** | **99.54** | 28.7 | 20.8 | **1084** | **0.37** | **4.6** |
| Budget = 30% | R:Replace | 92.19 | 98.26 | 29 | 21 | 4070.6 | 1.37 | 2 |
| | R:Majority Vote | 94.91 | 99.02 | 27.3 | 19.4 | 2346.6 | 0.79 | 2.5 |
| | E:Replace | 91.75 | 98.14 | **29** | **21** | 4316 | 1.45 | 1.9 |
| | E:Majority Vote | 93.54 | 98.76 | 28.2 | 20.3 | 2970 | 1 | 2.3 |
| | E:MB | 96.24 | 99.35 | 27.8 | 19.9 | 1565 | 0.53 | 3.4 |
| | E:MBME | 96.64 | 99.48 | 26.9 | 18.9 | 1280 | 0.43 | 3.7 |
| | E:CR 2 | 97.42 | 99.55 | 27.7 | 19.7 | 1074 | 0.36 | 4.4 |
| | E:CR 5 | 98.43 | 99.71 | 27.4 | 19.4 | 689 | 0.23 | 6.2 |
| | E:CR 10 | **98.91** | **99.79** | 27.1 | 19.2 | **504** | **0.17** | **8.1** |
| Budget = 10% | R:Replace | 97.49 | 99.6 | 22.1 | 14.2 | 996.6 | 0.34 | 3.6 |
| | R:Majority Vote | 97.86 | 99.68 | 21.6 | 13.6 | 808.8 | 0.27 | 4.1 |
| | E:Replace | 97.5 | 99.6 | **23.7** | **15.7** | 996 | 0.34 | 3.9 |
| | E:Majority Vote | 97.5 | 99.6 | **23.7** | **15.7** | 996 | 0.34 | 3.9 |
| | E:MB | 98.08 | 99.72 | 22.7 | 14.8 | 696 | 0.23 | 4.9 |
| | E:MBME | 98.08 | 99.72 | 22.7 | 14.8 | 696 | 0.23 | 4.9 |
| | E:CR 2 | 98.68 | 99.83 | 20.7 | 12.8 | 427 | 0.14 | 6.6 |
| | E:CR 5 | 99.32 | 99.92 | 18.2 | 10.3 | 197 | 0.07 | 10.7 |
| | E:CR 10 | **99.57** | **99.95** | 17.2 | 9.2 | **122** | **0.04** | **15** |

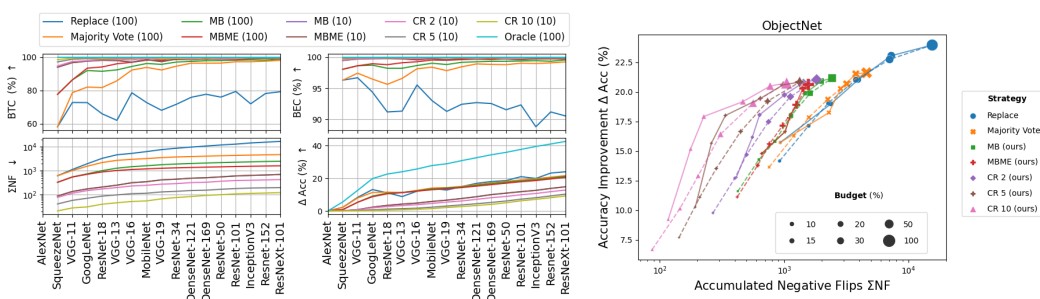

Figure 3: **Left:** Temporal evolution for ImageNet → ObjectNet over $T = 16$ prediction-update steps for a subset of strategies and budgets. **Right:** Comparison of all strategies after $T = 16$ on ObjectNet.

our Bayesian strategies clearly dominate the non-probabilistic baselines both in terms of accuracy and flip metrics, as can be seen from Tab. 1 and Fig. 2 (right). Moreover, the CR strategy appears to provide control over the number of negative flips via its cost-ratio hyperparameter without adversely affecting final accuracy across a range of budgets, as already observed for a budget of 10% in Fig. 2. Interestingly, the update rules seem to be optimal when evaluating on less than 100% budget. We attribute this to posterior approximation errors on ImageNet, which is being supported by extensive ablations in the supplement. Regarding backward compatibility (our ultimate goal), we find that **BTC** *and* **BEC** scores reliably outperform the baselines across all budgets. In particular, the CR 10 strategy seems to be especially suitable with scores close to 100%, i.e., oracle performance.

**Summary** Our method appears to successfully fulfill the three desiderata for backward-compatible prediction-updates in an i.i.d. setting. In particular, our CR strategy seems like the most promising candidate to (i) maintain high accuracy gains and (ii) introduce very few negative flips, when (iii) given only a small compute budget for re-evaluations.

## 6.3 Results for ImageNet → ObjectNet

Results for prediction-updates on ObjectNet are presented (similarly to § 6.2) in Fig. 3 and Tab. 1 (right). This transfer setting constitutes a much more challenging task. Nevertheless, we observe very similar behaviour to that discussed in § 6.2 and thus only point out the main differences. First, we note that - despite the smaller target data set - the difference in negative flips across different strategies and budgets is even larger on ObjectNet. For example, we observe a reduction in $\Sigma$ NF of more than two orders of magnitude between Replace (100) and CR (10), and about one order when comparing the two for the same budget. At the same time, differences in accuracy across strategies are also slightly more pronounced, especially for the smallest budget of 10%. Here, the more conservative CR strategies yield lower accuracy gains while MB and MBME maintain competetive accuracy gains. Our strategies are again clearly dominating in terms of backward compatibility w.r.t. BTC and BEC. We remark that these results are agnostic to any potential differences in the label space: they are based on a posterior over all 1000 ImageNet classes whereas ObjectNet only contains a subset of 113 of these classes.

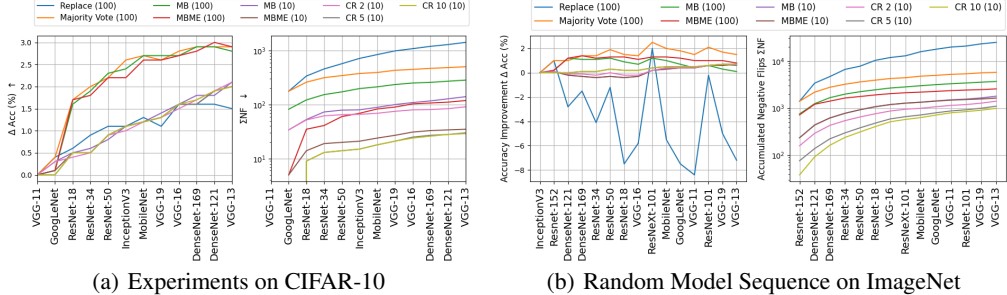

(a) Experiments on CIFAR-10          (b) Random Model Sequence on ImageNet

Figure 4: **Left:** Temporal evolution for CIFAR-10 over $T = 11$ prediction-update steps for a subset of strategies and budgets. Our approaches achieve competitive overall accuracy gains (even at 10% of the budget) while introducing only a fraction of negative flips. **Right:** Experiment on ImageNet with random model sequence suggesting robustness of our approach to situations where an ordering by performance of may not be available.

### 6.4 Further Experiments and Ablations

**Results for CIFAR-10** In Fig. 4 (a) we show the corresponding results on the CIFAR-10 dataset. Here, pre-trained models exhibit a higher level of accuracy ($\approx 93 - 95\%$) and we thus emulate an arguably more realistic scenario with models being released more frequently, thus with smaller accuracy differences. Our method shows very similar trends as we have worked out on ImageNet and ObjectNet. Interestingly, there is one novel characteristic: due to the presumably less steep increase in accuracy from one model to its successor and fewer class categories, we can form very accurate posterior beliefs which results in accuracy gains of all our methods that even outperform the accuracy-optimizing baselines concerning desideratum (i). A more comprehensive analysis of these experiments can be found in Appendix A.5.

**Role of the Selection Strategy** We also conduct a comparison between our entropy selection and the random baseline for all our methods across a range of budgets - see dashed lines in scatter plots. We find that random selection leads to substantially smaller accuracy gains, but also to fewer negative flips which is intuitive since random selection more often chooses "easy" samples for re-evaluation.

**Robustness to Random & Adversary Model Sequences** We have assumed that the models $C^t$ are *improving* over time. We thus also consider the scenario where $C^t$ arrive in a *random* or *adversarial* (i.e., strictly deteriorating) order. For the random order (see Fig. 4 (b)) we find that our methods - unlike, e.g., the replace strategy - achieve strict increases in accuracy while introducing much fewer negative flips. Even in the adversarial case, our methods improve accuracy during the initial steps with much fewer negative flips over the entire history. These findings suggest robustness of our approach to situations where an ordering by performance of $C^t$ may not be available. We refer to Appendix A.3 for more details.

**Reducing Re-Evaluations Matters at Scale** The re-evaluations of a sample using deep neural network based models clearly dominate computational cost as compared to our method. As an example, the forward pass using the public ImageNet PyTorch models takes up to 550 (biggest VGG and ResNets) times longer than the unoptimized implementation of our method backbone. We analyze this in more detail in Appendix A.9. For very large data sets and with new models generally increasing in size, reducing the inference budget B is of crucial importance, emphasizing the relevance of desideratum 3.

## 7 Conclusion

The Prediction Update Problem appears frequently in practice and can take different forms. It relates to many different subfields of ML that we have discussed in § 4 and § 5, and there are interesting extensions (structured prediction, adaptive budgets) and improvements (modeling data set structure, across-dataset similarity, domain adaptation, calibration techniques) that need to be worked out. In this work, we have studied the classification case and proposed a Bayesian update rule based on simple assumptions. Empirically, we find improvements along the dimensions we set out to achieve, and we hope that progress on this problem will democratize ML usage even further as it lowers the bar for benefitting from the tremendous progress in model design seen over the last years.

## Acknowledgements

The authors would like to thank Matthias Bethge, Thomas Brox, Chris Russell, Karsten Roth, Nasim Rahaman, Alex Smola, Yuanjun Xiong and Stefano Soatto for helpful discussions and feedback.

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
