# A Supplementary Material

In this supplement, we present additional results and ablations of experiments referred to in the main paper.

## A.1 Assumption of Conditionally-Independent Classifiers

In the case of two given classifiers our assumption implies that $\forall n, \forall (t, t') : I(\hat{Y}_n^t; \hat{Y}_n^{t'} | Y_n) = 0$. To quantify deviations of Assumption 1, one could therefore measure this conditional mutual information, but note that this in itself is challenging on finite data and would again inherit model assumptions through the choice of the estimator. Since pairwise independence does not imply mutual independence, even having these quantities might not be sufficient for a long classifier sequence. An alternative approach could thus be to measure the Maximum Mean Discrepancy (MMD) between the LHS and RHS of Assumption 1, which can be viewed as a conditional version of the d-dimensional Hilbert Schmidt Independence Criterion (dHSIC), see [30].

Importantly, all of the above approaches aim to quantify and subsequently incorporate correlations between classifier predictions, and thus rely on having large amounts of labelled data to properly estimate them which is almost never the case in practice. Since our goal was to devise an approach that can be useful in practice, when the number of classes, observations, and models is typically very large, we believe that the model choice behind Assumption 1 is reasonable; it enables both tractable inference and parameter estimation.

## A.2 Backward Compatibility Scores

For the reported BTC and BEC scores we follow the definition proposed in [43] with $h_1$ and $h_2$ being the predictors at time step 1 and time step 2. Translated to our setup, we associate $h_1$ with our estimated and stored prediction at time step 1 whereas $h_2$ corresponds to the updated prediction dataset at time step 2:

**Backward Trust Compatibility (BTC) score.** "The ratio of points in a held-out test set (e.g., $D_{test}$) that predicted correctly among all points $h_1$ had already predicted correctly." [43]

$$\text{BTC} = \frac{\sum_{i=1}^{|D|} \mathbf{1}[h_1(x_i) = y_i, h_2(x_i) = y_i]}{\sum_{i=1}^{|D|} \mathbf{1}[h_1(x_i) = y_i]} \tag{4}$$

**Backward Error Compatibility (BEC) score.** "The proportion of points in a held-out test set that $h_2$ predicted incorrectly, out of which $h_1$ also predicted incorrectly, thus capturing the probability that a mistake made by $h_2$ is not new." [43]

$$\text{BEC} = \frac{\sum_{i=1}^{|D|} \mathbf{1}[h_1(x_i) \neq y_i, h_2(x_i) \neq y_i]}{\sum_{i=1}^{|D|} \mathbf{1}[h_2(x_i) \neq y_i]} \tag{5}$$

## A.3 Robustness to Random and Adversarial Model Sequences

As already eluded to in § 6.4, in our main experiments we have assumed that the models $C^t$ are *improving* over time. In general, this assumption may be justified by having observed superior performance of a new model for the source domain on which they were trained, prior to deciding to use it for re-evaluation on the target data set. However, such information may not always be available.

To check for robustness against violations of the assumption of improving classifiers, we also consider scenarios where $C^t$ arrive in a *random* or *adversarial* (i.e., strictly deteriorating) order with respect to their accuracy. To avoid unrealistically large fluctuations in the accuracy of new classifiers, we removed AlexNet and SqueezeNet as the most poorly performing models from the *random* model sequence for this set of experiments. Results on ImageNet in the form of the temporal evolution of accuracy improvement and accumulated negative flips across different strategies and budgets are shown in Fig. 5.

For the random ordering in Fig. 5 (a), we find that our methods—unlike, e.g., the replace strategy—achieve strict increases in *final* accuracy while introducing much fewer negative flips, e.g., almost an order of magnitude fewer for our CR strategies, compared to the Majority Vote baseline. Even in the adversarial case in Fig. 5 (b), our methods improve accuracy during the initial steps and introduce much fewer negative flips over the entire history. These findings suggest robustness of our approach to situations where an ordering by performance of $C^t$ may not be available.

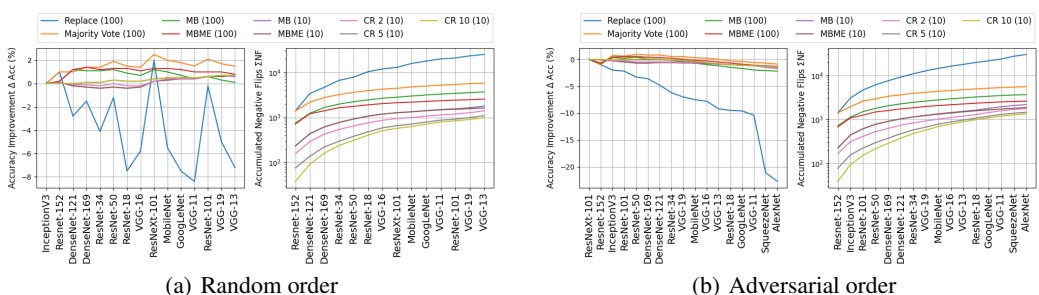

(a) Random order                                    (b) Adversarial order

Figure 5: Temporal evolution of accuracy improvement and accumulated negative flips on ImageNet for two scenarios where the models $C^t$ do not exhibit improving performance, but instead arrive in (a) random or (b) adversarial order. As can be seen, our methods are robust in both these cases while introducing much fewer negative flips than the non-probabilistic baselines.

## A.4    Role of Confusion Matrix Estimates

For the results of our main experiments reported in § 6.2 and § 6.3, we estimated only the diagonal elements of the confusion matrices and set the off-diagonal elements to a constant, c.f. § 3.1. To investigate the role of the estimation procedure for the confusion matrices $\pi^t$, we also report results for the case where we estimate the *full* confusion matrix of each classifier (i.e., including the off-diagonal elements). To avoid off-diagonal elements which are estimated to be zero due to the large number of classes and the limited size of the validation set split, we use a Laplace-smoothed version of the maximum likelihood estimate, i.e., we add a one to each count and normalise accordingly, as suggested in § 4.

Results of this comparison between different confusion matrix estimators for ImageNet and ObjectNet are shown in Fig. 6. (The corresponding results for CIFAR-10 are shown in Fig. 7 and are discussed in more detail in A.5.) As is apparent from the comparison on ImageNet and ObjectNet in Fig. 6, we indeed observe generally improved performance from estimating full confusion matrices, as speculated in § 4.

In terms of final accuracy, we observe a different behaviour across different budgets. For large budgets, estimating full confusion matrices leads to substantial accuracy gains on ImageNet and to roughly equal or slightly improved accuracy on ObjectNet. For small budgets, on the other hand, we find similar or smaller accuracy gains for full confusion matrix estimates; importantly, this effect is most pronounced for the more conservative CR prediction-update strategies. In terms of negative flips, we observe slight to major reductions across all budgets and strategies with full confusion matrix estimates compared to only estimating the diagonals, i.e., the class-specific accuracies.

We believe that this behaviour is intuitive and can be explained as follows. Estimating the full confusion matrices with smoothed counts generally yields smaller diagonal elements and larger off-diagonal elements. This, in turn, means that our posterior beliefs (in particular, the ratio between two consecutive MAP values) change less drastically after new re-evaluations, so that the more conservative CR strategies update fewer labels and thus experience smaller and slower accuracy gains when only few samples can be re-evaluated. At the same time, the resulting posterior estimates are likely more accurate which is consistent with larger gains for large budgets and the reduced number of negative flips. In summary, better confusion matrix estimates result in a further reduction in the number of negative flips across the board. For small budgets and conservative prediction-update strategies, however, this comes with a trade-off in overall accuracy gain.

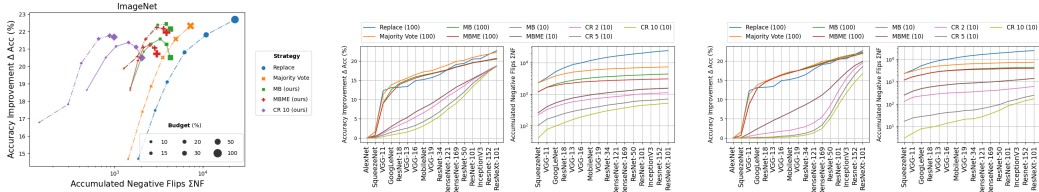

(a) ImageNet: Role of Confusion Matrix

(b) ImageNet: Diagonal Confusion Matrix

(c) ImageNet: Full Confusion Matrix

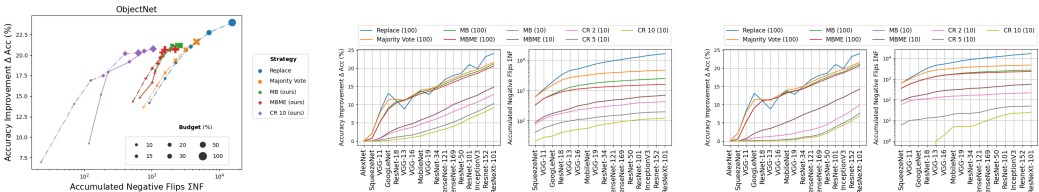

(d) ObjectNet: Role of Confusion Matrix

(e) ObjectNet: Diagonal Confusion Matrix

(f) ObjectNet: Full Confusion Matrix

Figure 6: Investigation into the role of different confusion matrix estimates on ImageNet (top) and ObjectNet (bottom). Dashed lines in the scatter plots (a) and (d) represent results for when the full confusion matrices (i.e., including the off-diagonals) are estimated from smoothed counts on a split of the validation set, and solid lines represent results under the less accurate confusion matrix estimates where only diagonal elements (i.e., the class-specific accuracies) are estimated, as in the experiments of the main paper.

## A.5 Experiments on CIFAR-10

CIFAR-10 is an image classification dataset comprising images of objects from 10 distinct classes (airplanes, cars, birds, cats, deer, dogs, frogs, horses, ships, trucks). It contains a training set of 50,000 images and a test set of size 10,000. It is thus a rather simple classification dataset compared to ImageNet and ObjectNet but has been a key driver for advancing ML models and computer vision before the introduction of ImageNet. Using the CIFAR-10 test set as our target dataset, we can study the Prediction Updates Problem in an i.i.d. setting with a small number of classes.

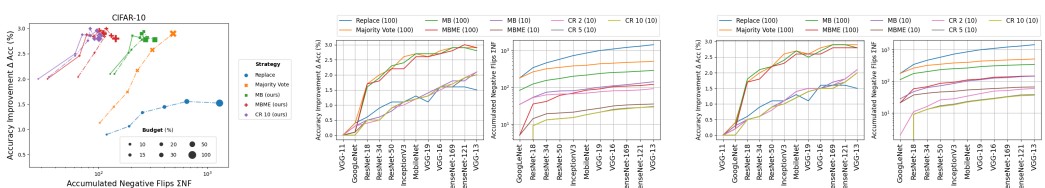

(a) Comparison of final performance

(b) Diagonal confusion matrix estimate

(c) Full confusion matrix estimate

Figure 7: Summary of prediction-updates on CIFAR-10. Similar to Fig. 6, dashed lines in (a) represent results with full confusion matrices estimated from smoothed counts and solid lines represent using a less accurate confusion matrix estimate where only diagonals are estimated from data. We observe very similar behaviour to that described for ImageNet and ObjectNet in § 6.2 and § 6.3, respectively. The most interesting difference compared to our experiments on ImageNet and ObjectNet, is that, on CIFAR-10, our methods achieve substantially larger accuracy gains compared with the baselines, even for small budgets.

Similar to our experiments on ImageNet and ObjectNet reported in § 6 where we used models which had been pre-trained on ImageNet,[6] for our experiments on CIFAR-10, we use models which have instead been pre-trained on CIFAR-10, available from an open source github repository.[7] This repository contains pre-trained models for a subset of the same architectures listed in § 6.1, excluding

---

[6]ImageNet pretrained models: link

[7]CIFAR-10 pretrained models: link

AlexNet, SqueezeNet, ResNet-101, ResNeXt; the released VGG pre-trained models represent variants with batch normalization. We order these CIFAR-10 pre-trained models according to their validation accuracies reported in the above repository. Compared to ImageNet and ObjectNet, pre-trained models on CIFAR-10 exhibit a much higher level of accuracy (between 93% and 95%) with much smaller differences between the best and worst model. As a consequence, our experiment on CIFAR-10 emulates a scenario in which incoming new classifiers exhibit smaller performance gains. This might reflect a practical prediction-updates setting with a higher frequency of incoming new models.

The results of our experiments on CIFAR-10 are summarised in Fig. 7. Our method shows very similar trends w.r.t. all update strategies and metrics as we have worked out on ImageNet and ObjectNet. Interestingly, there is one novel characteristic when it comes to the accuracy gains: due to the presumably less steep increase in accuracy from one model to its successor and fewer class categories, we can form more accurate posterior beliefs which results in accuracy gains for our CR 10 with smallest budget that outperforms the backfilling Replace (100) baseline. The overall accuracy gain similarly remains competitive with all Majority Vote variants while exhibiting only a fraction of the negative flips. Again, BTC and BEC strongly outperform all baselines along every update step for all our methods as can be seen in Fig. 8

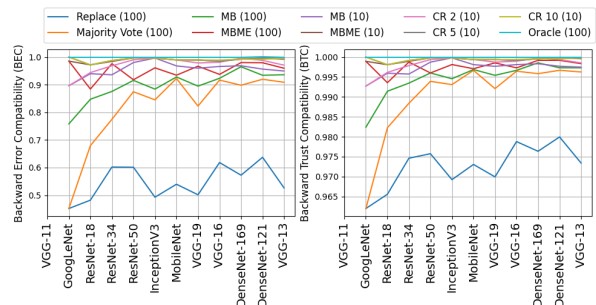

Figure 8: BTC and BEC evolution on CIFAR10 with utilizing diagonal confusion matrix elements only.

### A.6 On Peaking Accuracies

In Fig. 2 (right, solid lines), we observed a peak in $\Delta$**Acc** at 30% budget for all our strategies when performing prediction-updates on ImageNet. This suggests that our update rules are no longer beneficial on this dataset when re-evaluating more than the top 30% of samples with highest posterior entropy. From our ablation experiments on different confusion matrix estimates shown in Fig. 6 (a), we observe that this peaking phenomenon is much less pronounced and occurs only at larger budgets of $B \geq 50\%$ when estimating the full confusion matrix based on the half of the validation set on which we do not evaluate. We therefore conjecture that this behaviour may be related to inaccurate estimates of the posterior resulting from our approximation of the unknown confusion matrices (and possibly the assumption of conditionally independent classifiers).

To further investigate this hypothesis, we performed an ablation where we use the *full* validation set both to estimate the confusion matrix (with smoothing) and for evaluation—i.e., we do not use a 50-50 split of the validation set as in Fig. 6 and all experiment in the main paper—thus matching almost exactly[8] the confusion counts statistics of the evaluation set. As a result, we can expect our estimates of the posterior distribution to be much more accurate in this case.

The results are shown in Fig. 9 in dashed lines, compared to the peaking behaviour from our default method in solid lines for reference. Apart from substantial gains in overall accuracy, we find that the peaking phenomenon disappears almost entirely,with only a very slight drop in accuracy remaining between budgets of 50% and 100% for some strategies. This seems to confirm our hypothesis that the main source of the peaking behaviour are approximation errors in our posterior estimates resulting from inaccurate confusion matrices.

### A.7 Role of the Selection Strategy

We also conduct an ablation study on the role of the selection strategy for samples to be re-evaluated as was being mentioned in § 6.4. In particular we conduct a comparison between our posterior entropy selection and the random selection (without replacement) baseline for all our methods across a range of budgets—see Fig. 10 for these ablation results on all three datasets CIFAR-10, ImageNet and

---

[8]up to the one count smoothing effect.

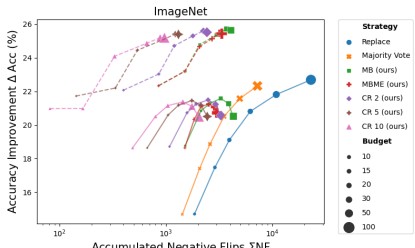

Figure 9: Further investigation into the peaking phenomenon observed for prediction-updates on ImageNet in Fig. 2 (right). Solid lines are as in Fig. 2 (right), i.e., using a 50:50 split of the validation set, using one half to estimate the diagonal elements of the confusion matrices and the other half as our target evaluation set; dashed lines show the results of our ablation experiment where we use the full validation set to both estimate the exact statistics of the confusion matrix and to evaluate the different methods. Since the evaluation set for the latter is twice as large, we scaled the reported negative flips by 0.5 for ease of comparison.

ObjectNet. Along all datasets, we find that random selection leads to substantially smaller accuracy gains, but also results in fewer negative flips. We can explain these fewer negative flips by the fact that our random selection more often chooses "easy" samples for re-evaluation. Under a random selection criteria, there is no preference in re-evaluating more "controversial" samples w.r.t previous predictions. This effect becomes particularly pronounced for small budgets. Our finding suggests that the selection strategy is indeed a relevant component, with room for improvement as discussed in § 4. This is also highlighting the potential control capability of this component regarding our three desiderata.

## A.8    Evolution of the Stored Label Distribution

For additional insights which samples are getting selected for re-evaluation by our entropy selection criterion, we also show some characteristic count distribution of correctly and incorrectly stored predictions along various update steps. See Fig. 11 for this distribution after every second update step on ImageNet using the MB strategy with a compute budget of $B = 10 \, (\%)$.

## A.9    Reducing re-evaluations matters at scale

To see that the re-evaluations of an example image using deep neural network based models clearly dominate computational cost as compared to our method backbone that involved computing the approximate posterior, label entropy and update strategy details, we measure the time for the plain model estimation vs the time for posterior update on the same Intel Xeon (Cascade Lake) CPU for the sake of comparison. We summarize our measurement in Tab. 2 for all used ImageNet models, showing the multiples in compute time required by the model inference (per image on average) compared to the timing per image of our fully unoptimized method implementation, i.e. the posterior update backbone. For very large data sets and with new models generally increasing in size, reducing the inference budget B is of crucial importance, emphasizing the relevance of desideratum 3.

## A.10    Result Tables of Main Experiments

For sake of completeness and reproducibility, we also provide a complete quantitative account of all the main experiments and ablations on all three datasets discussed throughout this paper in Tab. 3 and Tab. 4.

## A.11    On Utilizing Uncalibrated Soft Labels

Finally, we tested our method on incorporating the uncalibrated soft labels provided as output in the pre-trained ImageNet models by repeating the ImageNet and ObjectNet experiment (in combination with full confusion matrix estimation). As already discussed in § 4, deep neural networks are known to have unreliable uncertainty estimates [14, 25, 26, 47] and we therefore do not expect these

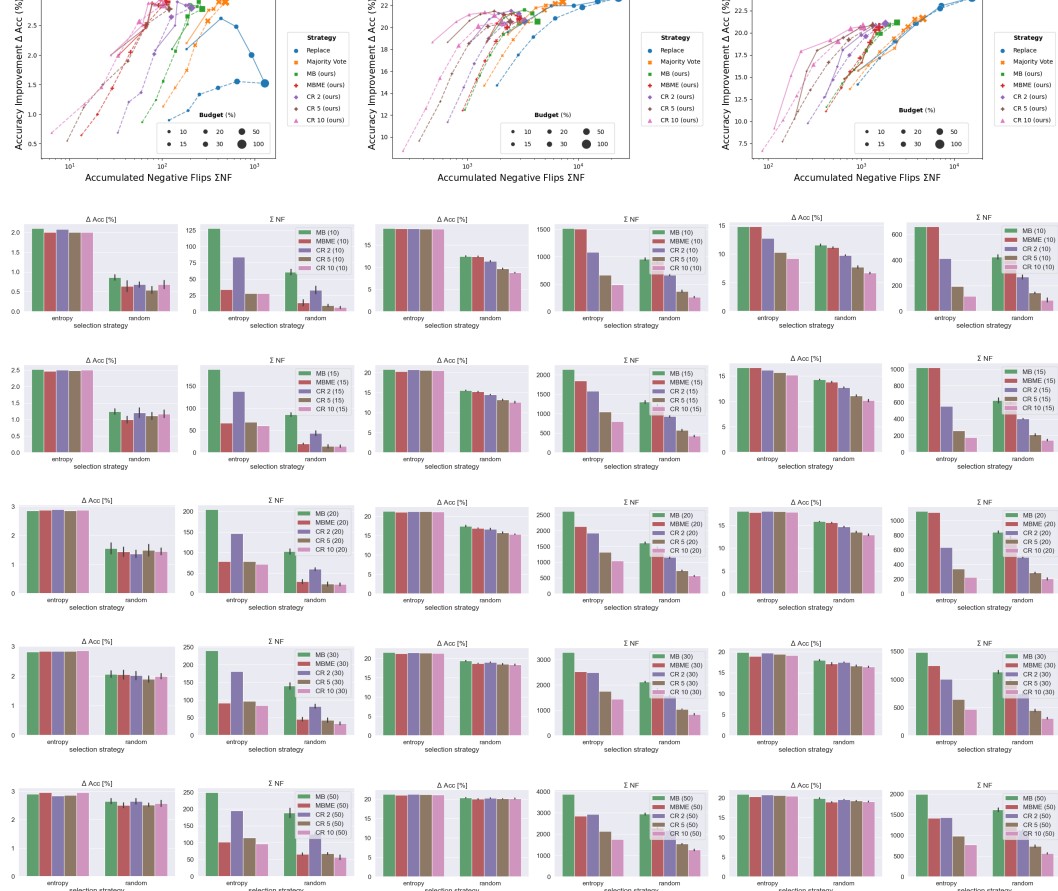

Figure 10: Ablation results for comparing random (dashed lines in scatter plots) and entropy-ranked (solid lines in scatter plots) selection strategies on CIFAR-10 (left column), ImageNet (middle column) and ObjectNet (right column). Different rows of the bar plots correspond to, from top to bottom, $B =$10, 15, 20, 30, 50%. Note that at 100% budgets, the two selection strategies are identical as all samples are re-evaluated at each step. We find that random selection leads to substantially smaller accuracy gains, but also fewer negative flips. This is intuitive as random selection more often chooses "easy" samples for re-evaluation. The effect is particularly pronounced for small budgets.

Table 2: Time measurements of a single forward pass versus the mean average time to compute the posterior update (0.406 milliseconds) on the same computational ressources for comparability. For reference we extrapolated how long it would take to infer the predictions from 1B images.

| Model | model re-evaluation [milliseconds] | ratio inference vs posterior update | inference for 1B images [days] |
|---|---|---|---|
| alexnet | 29,6 | 73 | 343 |
| squeezenet | 32,6 | 80 | 378 |
| vgg11 | 125,5 | 309 | 1452 |
| googlenet | 65,6 | 162 | 759 |
| resnet18 | 31,5 | 78 | 364 |
| vgg13 | 164,1 | 404 | 1899 |
| vgg16 | 191,3 | 471 | 2215 |
| mobilenet | 23,1 | 57 | 267 |
| vgg19 | 223,0 | 550 | 2582 |
| resnet34 | 50,8 | 125 | 587 |
| densenet121 | 74,4 | 183 | 862 |
| densenet169 | 93,5 | 230 | 1082 |
| resent50 | 70,6 | 174 | 818 |
| resnet101 | 114,0 | 281 | 1319 |
| inceptionv3 | 115,8 | 285 | 1340 |
| resnet152 | 158,2 | 390 | 1831 |
| resnext101 32x8d | 206,6 | 509 | 2391 |

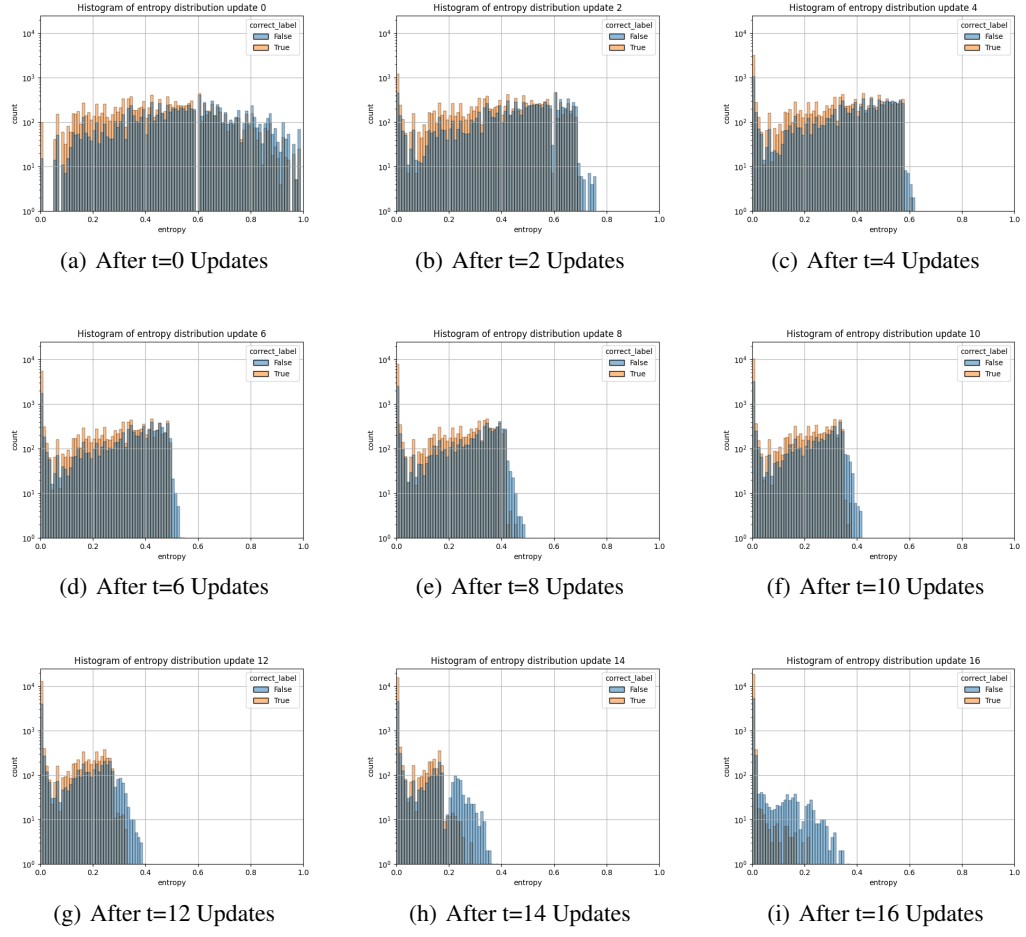

Figure 11: Count distribution evolution of correctly and incorrectly stored predictions on ImageNet using the MB update strategy under a budget of B = 10. Samples with currently stored predictions being false are depicted with blue, and correctly stored predictions with orange. We observe that under this MB strategy, samples with highest entropy tend to store false predictions, thus benefitting from re-evaluation particularly.

results to be representative of what our method could achieve if it has access to truly calibrated soft labels. We now utilize the full output softmax vector $\mathbf{p}^{sl} = C^t(\mathbf{x}_n)$ and refine our likelihood estimate in eq. (2) by multiplying the confusion matrix coefficients with the soft label vector, i.e. $\pi^t(\hat{y}_n^t, k) \rightarrow \sum_i p_i^{sl} \pi^t(i, k)$. We show the temporal evolution compared to the hard-label implementation using only diagonal confusion matrix estimates in Fig. 12 and a full account over the final performance metrics and all selection strategies and budgets in Tab. 5 and Tab. 6. Overall the results are slightly less conclusive and less consistent, most likely due to the known uncalibrated soft labels, emphasizing the strength of our method when only having deterministic labels. Specifically, on CIFAR-10, there are additional accuracy gains but often we accumulate more negative flips and BTC, BEC are typically slightly worse. On ImageNet, accuracy gains are sometimes worse without finding a clear pattern when this is the case, but negative flips are sometimes significantly lower. BTC, BEC seem to be better on the majority of update strategies and labels. On ObjectNet, we likewise find no clear patter in the accuracy gains. However, accumulated negative flips are generally much lower and both, BTC and BEC, even higher (very close to 1.0) but we want to emphasize that they were already very high for the ObjectNet experiments from the main paper.

Table 3: Full results of all experiments on **ImageNet (left) and ObjectNet (right)** under the standard setting of an **improving sequence of models** and **estimating confusion matrices on a separate split** of the data (i.e., not on the target data set) as explained in § 6.1. The first number in each cell refers to estimating only the diagonal elements (i.e., the class-specific accuracies) of confusion matrices, and the **second number (in brackets) refers to estimating the full confusion matrices** with smoothing. The character **E** in front of a budget indicates that selection for re-evaluation is based on our posterior label **entropy criterion** (e.g., E30 refers to selecting the top 30% samples with highest entropy), and the character **R** indicates selecting a **randomly** sampled subset without replacement. Note that entropy-based selection requires a posterior and is thus not applicable for the baselines Replace and Majority Vote.

**ImageNet (left)**

| Strategy Budget % | Acc (%) | ΔAcc (%) | Σ NF | NFR (%) | PF / NF | Avg. BTC | Avg. BEC |
|---|---|---|---|---|---|---|---|
| 100:Oracle | 91.2 (91.2) | 34.7 (34.7) | 0 (0) | 0.0 (0.0) | NaN (NaN) | 100.0 (100.0) | 100.0 (100.0) |
| R100:Replace | 79.2 (79.2) | 22.7 (22.7) | 24214 (24214) | 6.05 (6.05) | 1.2 (1.2) | 91.37 (91.37) | 77.71 (77.71) |
| R100:Majority Vote | 78.9 (78.9) | 22.3 (22.3) | 7352 (7352) | 1.84 (1.84) | 1.8 (1.8) | 97.18 (97.18) | 93.95 (93.95) |
| E100:MB | 77.1 (78.7) | 20.5 (22.2) | 4378 (4373) | 1.09 (1.09) | 2.2 (2.3) | 98.32 (98.32) | 96.45 (96.4) |
| E100:MBME | 77.3 (78.5) | 20.7 (22.0) | 3057 (3910) | 0.76 (0.98) | 2.7 (2.4) | 98.78 (98.48) | 97.69 (96.88) |
| E100:CR 2 | 77.1 (78.5) | 20.6 (22.0) | 3368 (2342) | 0.84 (0.59) | 2.5 (3.3) | 98.72 (99.12) | 97.19 (98.0) |
| E100:CR 5 | 77.1 (78.4) | 20.5 (21.8) | 2520 (1300) | 0.63 (0.33) | 3.0 (5.2) | 99.06 (99.52) | 97.82 (98.86) |
| E100:CR 10 | 77.0 (78.2) | 20.5 (21.7) | 2112 (1023) | 0.53 (0.26) | 3.4 (6.3) | 99.22 (99.62) | 98.15 (99.11) |
| R50:Replace | 78.4 (78.4) | 21.8 (21.8) | 11478.8 (11478.8) | 2.87 (2.87) | 1.5 (1.5) | 95.82 (95.82) | 89.88 (89.88) |
| R50:Majority Vote | 78.1 (78.1) | 21.6 (21.6) | 5048.2 (5048.2) | 1.26 (1.26) | 2.1 (2.1) | 98.07 (98.07) | 95.92 (95.92) |
| R50:MB | 76.9 (77.7) | 20.4 (21.1) | 3022.8 (3052.8) | 0.76 (0.76) | 2.7 (2.7) | 98.83 (98.82) | 97.62 (97.57) |
| R50:MBME | 76.5 (77.5) | 20.0 (20.9) | 2325.4 (2917.6) | 0.58 (0.73) | 3.1 (2.8) | 99.08 (98.87) | 98.27 (97.72) |
| R50:CR 2 | 76.8 (77.4) | 20.3 (20.8) | 2233.6 (1572.8) | 0.56 (0.39) | 3.3 (4.3) | 99.14 (99.4) | 98.22 (98.74) |
| R50:CR 5 | 76.6 (77.0) | 20.1 (20.4) | 1603.8 (840.6) | 0.4 (0.21) | 4.1 (7.1) | 99.39 (99.68) | 98.69 (99.32) |
| R50:CR 10 | 76.7 (76.7) | 20.1 (20.2) | 1314.6 (659.2) | 0.33 (0.16) | 4.8 (8.6) | 99.51 (99.75) | 98.92 (99.46) |
| E50:Replace | 78.9 (79.6) | 22.4 (23.1) | 15732 (15503) | 3.93 (3.88) | 1.4 (1.4) | 94.41 (94.53) | 85.37 (85.23) |
| E50:Majority Vote | 78.7 (79.1) | 22.2 (22.6) | 6318 (6420) | 1.58 (1.6) | 1.9 (1.9) | 97.6 (97.58) | 94.74 (94.55) |
| E50:MB | 77.8 (79.0) | 21.3 (22.4) | 3969 (3941) | 0.99 (0.99) | 2.3 (2.4) | 98.48 (98.5) | 96.78 (96.71) |
| E50:MBME | 77.6 (78.7) | 21.1 (22.2) | 2904 (3636) | 0.73 (0.91) | 2.8 (2.5) | 98.85 (98.6) | 97.79 (97.05) |
| E50:CR 2 | 77.8 (78.8) | 21.2 (22.2) | 3014 (2128) | 0.75 (0.53) | 2.8 (3.6) | 98.86 (99.21) | 97.5 (98.17) |
| E50:CR 5 | 77.7 (78.6) | 21.2 (22.0) | 2214 (1168) | 0.55 (0.29) | 3.4 (5.7) | 99.18 (99.57) | 98.09 (98.96) |
| E50:CR 10 | 77.8 (78.3) | 21.1 (21.8) | 1832 (913) | 0.46 (0.23) | 3.9 (7.0) | 99.33 (99.67) | 98.4 (99.19) |
| R30:Replace | 77.4 (77.4) | 20.8 (20.8) | 6546.4 (6546.4) | 1.64 (1.64) | 1.8 (1.8) | 97.56 (97.56) | 94.53 (94.53) |
| R30:Majority Vote | 77.1 (77.1) | 20.5 (20.5) | 3616.4 (3616.4) | 0.9 (0.9) | 2.4 (2.4) | 98.6 (98.6) | 97.18 (97.18) |
| R30:MB | 75.9 (76.3) | 19.4 (19.7) | 2186.2 (2200.4) | 0.55 (0.55) | 3.2 (3.2) | 99.14 (99.14) | 98.35 (98.34) |
| R30:MBME | 75.3 (76.2) | 18.7 (19.6) | 1859.6 (2183.8) | 0.46 (0.55) | 3.5 (3.2) | 99.26 (99.14) | 98.65 (98.36) |
| R30:CR 2 | 75.5 (75.5) | 19.0 (18.9) | 1607.0 (1092.4) | 0.4 (0.27) | 4.0 (5.3) | 99.37 (99.57) | 98.8 (99.19) |
| R30:CR 5 | 75.1 (74.7) | 18.5 (18.2) | 1087.2 (564.8) | 0.27 (0.14) | 5.3 (9.1) | 99.58 (99.78) | 99.17 (99.57) |
| R30:CR 10 | 74.9 (74.2) | 18.4 (17.7) | 875.4 (434.2) | 0.22 (0.11) | 6.2 (11.2) | 99.66 (99.83) | 99.33 (99.68) |
| E30:Replace | 78.5 (79.3) | 22.0 (22.8) | 9708 (8900) | 2.43 (2.23) | 1.6 (1.6) | 96.53 (96.82) | 91.01 (91.69) |
| E30:Majority Vote | 78.5 (79.2) | 22.0 (22.6) | 5232 (4989) | 1.31 (1.25) | 2.1 (2.1) | 98.03 (98.12) | 95.63 (95.79) |
| E30:MB | 78.1 (78.9) | 21.6 (22.4) | 3375 (3180) | 0.84 (0.8) | 2.6 (2.8) | 98.71 (98.79) | 97.25 (97.37) |
| E30:MBME | 77.8 (78.8) | 21.2 (22.2) | 2577 (3071) | 0.64 (0.77) | 3.1 (2.8) | 98.98 (98.83) | 98.04 (97.49) |
| E30:CR 2 | 78.0 (78.7) | 21.5 (22.1) | 2578 (1664) | 0.64 (0.42) | 3.1 (4.3) | 99.02 (99.37) | 97.86 (98.58) |
| E30:CR 5 | 78.0 (78.3) | 21.5 (21.8) | 1831 (890) | 0.46 (0.22) | 3.9 (7.1) | 99.32 (99.67) | 98.43 (99.21) |
| E30:CR 10 | 77.9 (78.1) | 21.4 (21.6) | 1517 (707) | 0.38 (0.18) | 4.5 (8.6) | 99.44 (99.74) | 98.67 (99.37) |
| R20:Replace | 75.7 (75.7) | 19.1 (19.1) | 4171.0 (4171.0) | 1.04 (1.04) | 2.1 (2.1) | 98.41 (98.41) | 96.71 (96.71) |
| R20:Majority Vote | 75.4 (75.4) | 18.9 (18.9) | 2690.2 (2690.2) | 0.67 (0.67) | 2.8 (2.8) | 98.95 (98.95) | 97.99 (97.99) |
| R20:MB | 74.0 (74.2) | 17.5 (17.6) | 1662.8 (1655.4) | 0.42 (0.41) | 3.6 (3.7) | 99.34 (99.34) | 98.81 (98.81) |
| R20:MBME | 73.5 (74.1) | 16.9 (17.6) | 1480.6 (1654.0) | 0.37 (0.41) | 3.9 (3.7) | 99.4 (99.34) | 98.97 (98.82) |
| R20:CR 2 | 73.2 (72.5) | 16.7 (15.9) | 1205.6 (786.2) | 0.3 (0.2) | 4.5 (6.1) | 99.52 (99.68) | 99.15 (99.45) |
| R20:CR 5 | 72.3 (71.3) | 15.8 (14.7) | 768.0 (388.8) | 0.19 (0.1) | 6.1 (10.5) | 99.69 (99.84) | 99.46 (99.73) |
| R20:CR 10 | 71.9 (70.5) | 15.4 (14.0) | 603.8 (280.2) | 0.15 (0.07) | 7.4 (13.5) | 99.76 (99.89) | 99.57 (99.81) |
| E20:Replace | 78.5 (79.0) | 22.0 (22.5) | 6191 (5430) | 1.55 (1.36) | 1.9 (2.0) | 97.74 (98.02) | 94.47 (95.16) |
| E20:Majority Vote | 78.3 (78.7) | 21.8 (22.1) | 4295 (3788) | 1.07 (0.95) | 2.3 (2.5) | 98.37 (98.56) | 96.46 (96.89) |
| E20:MB | 77.9 (78.1) | 21.3 (21.6) | 2700 (2371) | 0.68 (0.59) | 3.0 (3.3) | 98.96 (99.09) | 97.86 (98.11) |
| E20:MBME | 77.6 (78.1) | 21.1 (21.6) | 2183 (2371) | 0.55 (0.59) | 3.4 (3.3) | 99.13 (99.09) | 98.39 (98.11) |
| E20:CR 2 | 77.8 (77.8) | 21.3 (21.3) | 1999 (1197) | 0.5 (0.3) | 3.7 (5.4) | 99.23 (99.54) | 98.41 (99.04) |
| E20:CR 5 | 77.7 (77.4) | 21.2 (20.8) | 1383 (599) | 0.35 (0.15) | 4.8 (9.7) | 99.47 (99.77) | 98.87 (99.49) |
| E20:CR 10 | 77.7 (76.7) | 21.2 (20.2) | 1101 (461) | 0.28 (0.12) | 5.8 (11.9) | 99.58 (99.83) | 99.08 (99.62) |
| R10:Replace | 71.3 (71.3) | 14.7 (14.7) | 1958.4 (1958.4) | 0.49 (0.49) | 2.9 (2.9) | 99.22 (99.22) | 98.63 (98.63) |
| R10:Majority Vote | 71.2 (71.2) | 14.7 (14.7) | 1481.4 (1481.4) | 0.37 (0.37) | 3.5 (3.5) | 99.4 (99.4) | 98.98 (98.98) |
| R10:MB | 69.0 (69.0) | 12.5 (12.4) | 991.4 (963.6) | 0.25 (0.24) | 4.1 (4.2) | 99.59 (99.6) | 99.35 (99.36) |
| R10:MBME | 68.9 (69.0) | 12.4 (12.4) | 944.8 (976.8) | 0.24 (0.24) | 4.3 (4.2) | 99.61 (99.6) | 99.38 (99.35) |
| R10:CR 2 | 67.9 (66.0) | 11.4 (9.5) | 703.6 (432.2) | 0.18 (0.11) | 5.0 (6.5) | 99.71 (99.82) | 99.54 (99.73) |
| R10:CR 5 | 66.2 (64.2) | 9.6 (7.6) | 393.4 (147.0) | 0.1 (0.04) | 7.1 (14.0) | 99.84 (99.94) | 99.75 (99.91) |
| R10:CR 10 | 65.3 (63.5) | 8.8 (7.0) | 282.6 (104.8) | 0.07 (0.03) | 8.7 (17.6) | 99.88 (99.96) | 99.82 (99.93) |
| E10:Replace | 76.1 (77.3) | 19.5 (20.8) | 2468 (2320) | 0.62 (0.58) | 3.0 (3.2) | 99.04 (99.11) | 98.12 (98.12) |
| E10:Majority Vote | 75.9 (77.0) | 19.3 (20.5) | 2417 (2235) | 0.6 (0.56) | 3.0 (3.3) | 99.06 (99.14) | 98.18 (98.22) |
| E10:MB | 75.3 (76.4) | 18.8 (19.9) | 1557 (1394) | 0.39 (0.35) | 4.0 (4.6) | 99.38 (99.44) | 98.89 (98.98) |
| E10:MBME | 75.2 (76.4) | 18.7 (19.9) | 1533 (1394) | 0.38 (0.35) | 4.0 (4.6) | 99.38 (99.44) | 98.92 (98.98) |
| E10:CR 2 | 75.3 (76.1) | 18.7 (19.5) | 1118 (618) | 0.28 (0.15) | 5.2 (8.9) | 99.55 (99.75) | 99.22 (99.56) |
| E10:CR 5 | 75.2 (75.2) | 18.6 (18.6) | 700 (246) | 0.18 (0.06) | 7.7 (19.9) | 99.72 (99.9) | 99.51 (99.81) |
| E10:CR 10 | 75.2 (73.3) | 18.6 (16.8) | 515 (170) | 0.13 (0.04) | 10.1 (25.7) | 99.79 (99.93) | 99.64 (99.87) |

**ObjectNet (right)**

| Strategy (Budget %) | Acc (%) | ΔAcc (%) | Σ NF | NFR (%) | PF / NF | Avg. BTC | Avg. BEC |
|---|---|---|---|---|---|---|---|
| 100:Oracle | 50.5 (50.5) | 42.6 (42.6) | 0 (0) | 0.0 (0.0) | NaN (NaN) | 100.0 (100.0) | 99.99 (99.99) |
| R100:Replace | 31.9 (31.9) | 24.0 (24.0) | 16669 (16669) | 5.62 (5.62) | 1.3 (1.3) | 72.65 (72.65) | 92.61 (92.61) |
| R100:Majority Vote | 29.6 (29.6) | 21.6 (21.6) | 4690 (4690) | 1.58 (1.58) | 1.9 (1.9) | 89.99 (89.99) | 98.02 (98.02) |
| E100:MB | 29.1 (29.1) | 21.2 (21.1) | 2477 (2675) | 0.83 (0.9) | 2.6 (2.5) | 94.46 (94.04) | 98.96 (98.88) |
| E100:MBME | 28.6 (28.7) | 20.6 (20.7) | 1599 (2267) | 0.54 (0.76) | 3.4 (2.7) | 95.86 (94.63) | 99.34 (99.06) |
| E100:CR 2 | 29.0 (28.8) | 21.0 (20.9) | 1876 (1611) | 0.63 (0.54) | 3.1 (3.4) | 95.92 (96.54) | 99.21 (99.32) |
| E100:CR 5 | 28.8 (28.5) | 20.8 (20.6) | 1372 (853) | 0.46 (0.29) | 3.8 (5.5) | 97.18 (98.28) | 99.41 (99.63) |
| E100:CR 10 | 28.7 (28.2) | 20.8 (20.3) | 1084 (660) | 0.37 (0.22) | 4.6 (6.7) | 97.82 (98.68) | 99.54 (99.72) |
| R50:Replace | 30.7 (30.7) | 22.7 (22.7) | 7583.8 (7583.8) | 2.56 (2.56) | 1.6 (1.6) | 86.63 (86.63) | 96.69 (96.69) |
| R50:Majority Vote | 28.6 (28.6) | 20.7 (20.7) | 3281.6 (3281.6) | 1.11 (1.11) | 2.2 (2.2) | 93.01 (93.01) | 98.62 (98.62) |
| R50:MB | 27.8 (27.7) | 19.9 (19.8) | 1676.6 (1780.4) | 0.56 (0.6) | 3.2 (3.1) | 96.13 (95.9) | 99.3 (99.26) |
| R50:MBME | 26.9 (27.4) | 18.9 (19.4) | 1280.4 (1682.0) | 0.43 (0.57) | 3.7 (3.1) | 96.77 (96.04) | 99.48 (99.31) |
| R50:CR 2 | 27.5 (27.3) | 19.6 (19.4) | 1165.4 (923.0) | 0.39 (0.31) | 4.1 (4.9) | 97.3 (97.86) | 99.52 (99.62) |
| R50:CR 5 | 27.2 (26.6) | 19.2 (18.7) | 780.2 (469.6) | 0.26 (0.16) | 5.6 (8.4) | 98.26 (98.98) | 99.67 (99.8) |
| R50:CR 10 | 27.0 (26.2) | 19.1 (18.3) | 599.8 (353.2) | 0.2 (0.12) | 6.9 (10.6) | 98.7 (99.22) | 99.75 (99.85) |
| E50:Replace | 31.0 (31.3) | 23.1 (23.4) | 7882 (7730) | 2.66 (2.6) | 1.5 (1.6) | 86.26 (86.78) | 96.53 (96.59) |
| E50:Majority Vote | 29.5 (29.6) | 21.5 (21.7) | 3895 (3759) | 1.31 (1.27) | 2.0 (2.1) | 91.73 (92.2) | 98.36 (98.41) |
| E50:MB | 28.9 (29.1) | 20.9 (21.2) | 2079 (2081) | 0.7 (0.7) | 2.9 (2.9) | 95.23 (95.34) | 99.13 (99.13) |
| E50:MBME | 28.2 (28.6) | 20.3 (20.7) | 1452 (1944) | 0.49 (0.66) | 3.6 (3.0) | 96.18 (95.52) | 99.41 (99.19) |
| E50:CR 2 | 28.7 (28.7) | 20.8 (20.8) | 1506 (1136) | 0.51 (0.38) | 3.6 (4.4) | 96.64 (97.54) | 99.37 (99.52) |
| E50:CR 5 | 28.5 (28.4) | 20.6 (20.4) | 1049 (569) | 0.35 (0.19) | 4.6 (7.7) | 97.83 (98.88) | 99.55 (99.75) |
| E50:CR 10 | 28.4 (28.1) | 20.5 (20.2) | 828 (430) | 0.28 (0.14) | 5.6 (9.7) | 98.35 (99.16) | 99.64 (99.81) |
| R30:Replace | 29.0 (29.0) | 21.0 (21.0) | 4070.6 (4070.6) | 1.37 (1.37) | 2.0 (2.0) | 92.19 (92.19) | 98.26 (98.26) |
| R30:Majority Vote | 27.3 (27.3) | 19.4 (19.4) | 2346.6 (2346.6) | 0.79 (0.79) | 2.5 (2.5) | 94.91 (94.91) | 99.02 (99.02) |
| R30:MB | 25.9 (25.9) | 18.0 (17.9) | 1185.8 (1218.6) | 0.4 (0.41) | 3.8 (3.7) | 97.12 (97.01) | 99.52 (99.5) |
| R30:MBME | 25.1 (25.7) | 17.2 (17.8) | 1008.4 (1208.0) | 0.34 (0.41) | 4.2 (3.7) | 97.41 (97.03) | 99.59 (99.51) |
| R30:CR 2 | 25.4 (24.9) | 17.5 (16.9) | 782.2 (601.2) | 0.26 (0.2) | 5.2 (6.2) | 98.05 (98.46) | 99.68 (99.76) |
| R30:CR 5 | 24.6 (23.9) | 16.6 (16.0) | 476.2 (275.4) | 0.16 (0.09) | 7.5 (11.8) | 98.8 (99.32) | 99.81 (99.89) |
| R30:CR 10 | 24.4 (23.5) | 16.4 (15.5) | 331.6 (190.8) | 0.11 (0.06) | 10.2 (16.1) | 99.17 (99.53) | 99.86 (99.92) |
| E30:Replace | 29.0 (29.7) | 21.0 (21.7) | 4316 (4093) | 1.45 (1.38) | 1.9 (2.0) | 91.75 (92.39) | 98.14 (98.23) |
| E30:Majority Vote | 28.2 (28.6) | 20.3 (20.6) | 2970 (2703) | 1.0 (0.91) | 2.3 (2.4) | 93.54 (94.23) | 98.76 (98.87) |
| E30:MB | 27.8 (27.6) | 19.9 (19.7) | 1565 (1402) | 0.53 (0.47) | 3.4 (3.6) | 96.24 (96.66) | 99.35 (99.42) |
| E30:MBME | 26.9 (27.6) | 18.9 (19.7) | 1280 (1402) | 0.43 (0.47) | 3.7 (3.6) | 96.64 (96.66) | 99.48 (99.42) |
| E30:CR 2 | 27.7 (26.9) | 19.7 (19.0) | 1074 (675) | 0.36 (0.23) | 4.4 (6.2) | 97.42 (98.32) | 99.55 (99.72) |
| E30:CR 5 | 27.4 (26.0) | 19.4 (18.1) | 689 (312) | 0.23 (0.11) | 6.2 (11.8) | 98.43 (99.31) | 99.71 (99.87) |
| E30:CR 10 | 27.1 (25.4) | 19.2 (17.5) | 504 (214) | 0.17 (0.07) | 8.1 (16.2) | 98.91 (99.53) | 99.79 (99.91) |
| R20:Replace | 27.0 (27.0) | 19.1 (19.1) | 2453.8 (2453.8) | 0.83 (0.83) | 2.4 (2.4) | 94.82 (94.82) | 98.97 (98.97) |
| R20:Majority Vote | 25.8 (25.8) | 17.8 (17.8) | 1658.8 (1658.8) | 0.56 (0.56) | 3.0 (3.0) | 96.17 (96.17) | 99.32 (99.32) |
| R20:MB | 23.8 (23.7) | 15.8 (15.7) | 887.2 (892.2) | 0.3 (0.3) | 4.3 (4.3) | 97.74 (97.69) | 99.64 (99.64) |
| R20:MBME | 23.5 (23.9) | 15.6 (15.9) | 793.8 (866.2) | 0.27 (0.29) | 4.6 (4.4) | 97.93 (97.77) | 99.68 (99.65) |
| R20:CR 2 | 22.6 (21.8) | 14.7 (13.8) | 530.0 (392.2) | 0.18 (0.13) | 6.2 (7.6) | 98.57 (98.89) | 99.79 (99.84) |
| R20:CR 5 | 21.5 (20.4) | 13.5 (12.5) | 304.2 (171.2) | 0.1 (0.06) | 9.3 (14.5) | 99.14 (99.52) | 99.88 (99.93) |
| R20:CR 10 | 20.9 (19.9) | 12.9 (11.9) | 215.6 (121.4) | 0.07 (0.04) | 12.1 (19.2) | 99.4 (99.65) | 99.91 (99.95) |
| E20:Replace | 26.9 (27.5) | 19.0 (19.5) | 2588 (2480) | 0.87 (0.84) | 2.4 (2.5) | 94.54 (95.12) | 98.91 (98.94) |
| E20:Majority Vote | 26.2 (26.6) | 18.3 (18.7) | 2390 (2252) | 0.81 (0.76) | 2.4 (2.5) | 94.8 (95.41) | 99.0 (99.05) |
| E20:MB | 26.0 (26.3) | 18.1 (18.4) | 1169 (1075) | 0.39 (0.36) | 3.9 (4.2) | 97.01 (97.35) | 99.53 (99.56) |
| E20:MBME | 25.8 (26.3) | 17.8 (18.4) | 1131 (1075) | 0.38 (0.36) | 3.9 (4.2) | 97.06 (97.35) | 99.54 (99.56) |
| E20:CR 2 | 26.0 (26.2) | 18.1 (18.2) | 670 (490) | 0.23 (0.17) | 6.0 (7.9) | 98.14 (98.71) | 99.73 (99.8) |
| E20:CR 5 | 26.0 (25.6) | 18.0 (17.7) | 369 (193) | 0.12 (0.07) | 10.1 (18.0) | 98.97 (99.55) | 99.85 (99.92) |
| E20:CR 10 | 25.9 (24.9) | 17.9 (16.9) | 248 (134) | 0.08 (0.05) | 14.4 (24.4) | 99.34 (99.69) | 99.91 (99.94) |
| R10:Replace | 22.1 (22.1) | 14.2 (14.2) | 996.6 (996.6) | 0.34 (0.34) | 3.6 (3.6) | 97.49 (97.49) | 99.6 (99.6) |
| R10:Majority Vote | 21.6 (21.6) | 13.6 (13.6) | 808.8 (808.8) | 0.27 (0.27) | 4.1 (4.1) | 97.86 (97.86) | 99.68 (99.68) |
| R10:MB | 19.5 (19.3) | 11.6 (11.4) | 450.2 (442.4) | 0.15 (0.15) | 5.8 (5.8) | 98.7 (98.71) | 99.82 (99.83) |
| R10:MBME | 19.1 (19.1) | 11.1 (11.2) | 446.4 (449.0) | 0.15 (0.15) | 5.6 (5.6) | 98.69 (98.69) | 99.83 (99.82) |
| R10:CR 2 | 17.7 (16.1) | 9.8 (8.1) | 287.2 (196.6) | 0.1 (0.07) | 7.3 (8.7) | 99.11 (99.34) | 99.89 (99.92) |
| R10:CR 5 | 15.6 (14.1) | 7.7 (6.2) | 151.6 (74.6) | 0.05 (0.03) | 10.4 (16.4) | 99.5 (99.75) | 99.94 (99.97) |
| R10:CR 10 | 14.6 (13.5) | 6.7 (5.5) | 94.6 (46.0) | 0.03 (0.02) | 14.1 (23.4) | 99.68 (99.84) | 99.96 (99.98) |
| E10:Replace | 23.7 (25.7) | 15.7 (17.8) | 996 (849) | 0.34 (0.29) | 3.9 (4.9) | 97.5 (97.9) | 99.6 (99.65) |
| E10:Majority Vote | 23.7 (25.7) | 15.7 (17.7) | 996 (849) | 0.34 (0.29) | 3.9 (4.9) | 97.5 (97.9) | 99.6 (99.65) |
| E10:MB | 22.7 (22.3) | 14.8 (14.3) | 696 (560) | 0.23 (0.19) | 4.9 (5.8) | 98.08 (98.39) | 99.72 (99.78) |
| E10:MBME | 22.7 (22.3) | 14.8 (14.3) | 696 (560) | 0.23 (0.19) | 4.9 (5.8) | 98.08 (98.39) | 99.72 (99.78) |
| E10:CR 2 | 20.7 (17.8) | 12.8 (9.9) | 427 (213) | 0.14 (0.07) | 6.6 (9.6) | 98.68 (99.21) | 99.83 (99.92) |
| E10:CR 5 | 18.2 (15.7) | 10.3 (7.7) | 197 (49) | 0.07 (0.02) | 10.7 (30.3) | 99.32 (99.82) | 99.92 (99.98) |
| E10:CR 10 | 17.2 (14.9) | 9.2 (7.0) | 122 (24) | 0.04 (0.01) | 15.0 (54.9) | 99.57 (99.91) | 99.95 (99.99) |

Table 4: Full results of all experiments on **CIFAR10** under the standard setting of an **improving sequence of models** and **estimating confusion matrices on a separate split** of the data (i.e., not on the target data set) as explained in § 6.1. The first number in each cell refers to estimating only the diagonal elements (i.e., the class-specific accuracies) of confusion matrices, and the **second number (in brackets) refers to estimating the full confusion matrices** with smoothing. The character **E** in front of a budget indicates that selection for re-evaluation is based on our posterior label **entropy criterion** (e.g., E30 refers to selecting the top 30% samples with highest entropy), and the character **R** indicates selecting a **randomly** sampled subset without replacement. Note that entropy-based selection requires a posterior and is thus not applicable for the baselines Replace and Majority Vote.

| Strategy (Budget %) | Acc (%) | ΔAcc (%) | Σ NF | NFR (%) | PF / NF | Avg. BTC | Avg. BEC |
|---|---|---|---|---|---|---|---|
| 100:Oracle | 99.3 (99.3) | 6.1 (6.1) | 0 (0) | 0.0 (0.0) | NaN (NaN) | 100.0 (100.0) | 100.0 (100.0) |
| R100:Replace | 94.7 (94.7) | 1.5 (1.5) | 1418 (1418) | 2.58 (2.58) | 1.1 (1.1) | 97.26 (97.26) | 54.79 (54.79) |
| R100:Majority Vote | 96.1 (96.1) | 2.9 (2.9) | 502 (502) | 0.91 (0.91) | 1.3 (1.3) | 99.03 (99.03) | 81.94 (81.94) |
| E100:MB | 95.9 (95.9) | 2.8 (2.8) | 284 (340) | 0.52 (0.62) | 1.5 (1.4) | 99.45 (99.34) | 89.64 (87.71) |
| E100:MBME | 96.1 (96.0) | 2.9 (2.8) | 119 (146) | 0.22 (0.27) | 2.2 (2.0) | 99.77 (99.72) | 95.28 (94.47) |
| E100:CR 2 | 96.0 (96.0) | 2.8 (2.8) | 219 (170) | 0.4 (0.31) | 1.6 (1.8) | 99.58 (99.67) | 91.69 (93.28) |
| E100:CR 5 | 95.9 (95.9) | 2.8 (2.8) | 132 (126) | 0.24 (0.23) | 2.1 (2.1) | 99.75 (99.76) | 94.65 (94.98) |
| E100:CR 10 | 96.0 (96.0) | 2.9 (2.8) | 112 (103) | 0.2 (0.19) | 2.3 (2.4) | 99.79 (99.8) | 95.49 (95.92) |
| R50:Replace | 94.7 (94.7) | 1.6 (1.6) | 711.4 (711.4) | 1.29 (1.29) | 1.1 (1.1) | 98.62 (98.62) | 77.8 (77.8) |
| R50:Majority Vote | 95.7 (95.7) | 2.6 (2.6) | 327.4 (327.4) | 0.6 (0.6) | 1.4 (1.4) | 99.37 (99.37) | 89.06 (89.06) |
| R50:MB | 95.8 (95.8) | 2.7 (2.6) | 196.4 (227.8) | 0.36 (0.41) | 1.7 (1.6) | 99.62 (99.56) | 93.41 (92.47) |
| R50:MBME | 95.7 (95.6) | 2.5 (2.5) | 69.8 (90.6) | 0.13 (0.16) | 2.8 (2.4) | 99.87 (99.83) | 97.58 (96.93) |
| R50:CR 2 | 95.8 (95.8) | 2.6 (2.6) | 133.8 (104.2) | 0.24 (0.19) | 2.0 (2.3) | 99.74 (99.8) | 95.32 (96.24) |
| R50:CR 5 | 95.7 (95.7) | 2.5 (2.5) | 73.2 (69.0) | 0.13 (0.13) | 2.7 (2.8) | 99.86 (99.87) | 97.4 (97.59) |
| R50:CR 10 | 95.7 (95.7) | 2.6 (2.5) | 61.0 (60.6) | 0.11 (0.11) | 3.1 (3.1) | 99.88 (99.88) | 97.83 (97.85) |
| E50:Replace | 95.2 (95.2) | 2.0 (2.0) | 1019 (1012) | 1.85 (1.84) | 1.1 (1.1) | 98.04 (98.05) | 64.97 (64.69) |
| E50:Majority Vote | 96.1 (96.0) | 2.9 (2.9) | 428 (419) | 0.78 (0.76) | 1.3 (1.3) | 99.18 (99.19) | 84.48 (84.84) |
| E50:MB | 96.1 (96.0) | 2.9 (2.8) | 260 (289) | 0.47 (0.53) | 1.6 (1.5) | 99.5 (99.44) | 90.63 (89.8) |
| E50:MBME | 96.1 (96.0) | 3.0 (2.9) | 105 (134) | 0.19 (0.24) | 2.4 (2.1) | 99.8 (99.74) | 96.04 (95.09) |
| E50:CR 2 | 96.0 (96.0) | 2.8 (2.8) | 206 (149) | 0.37 (0.27) | 1.7 (1.9) | 99.6 (99.71) | 92.4 (94.27) |
| E50:CR 5 | 96.0 (95.9) | 2.9 (2.8) | 123 (112) | 0.22 (0.2) | 2.2 (2.2) | 99.76 (99.79) | 95.18 (95.67) |
| E50:CR 10 | 96.1 (96.0) | 3.0 (2.8) | 99 (95) | 0.18 (0.17) | 2.5 (2.5) | 99.81 (99.82) | 96.21 (96.35) |
| R30:Replace | 94.6 (94.6) | 1.4 (1.4) | 439.6 (439.6) | 0.8 (0.8) | 1.2 (1.2) | 99.15 (99.15) | 86.74 (86.74) |
| R30:Majority Vote | 95.3 (95.3) | 2.2 (2.2) | 238.4 (238.4) | 0.43 (0.43) | 1.5 (1.5) | 99.54 (99.54) | 92.54 (92.54) |
| R30:MB | 95.2 (95.3) | 2.1 (2.1) | 146.2 (163.2) | 0.27 (0.3) | 1.7 (1.7) | 99.72 (99.68) | 95.46 (94.91) |
| R30:MBME | 95.2 (95.2) | 2.0 (2.1) | 47.0 (64.2) | 0.09 (0.12) | 3.2 (2.6) | 99.91 (99.88) | 98.47 (97.92) |
| R30:CR 2 | 95.2 (95.2) | 2.0 (2.0) | 88.8 (64.6) | 0.16 (0.12) | 2.0 (2.4) | 99.83 (99.88) | 97.16 (97.92) |
| R30:CR 5 | 95.1 (95.0) | 1.9 (1.9) | 45.6 (47.0) | 0.08 (0.09) | 3.1 (3.0) | 99.91 (99.91) | 98.54 (98.49) |
| R30:CR 10 | 95.2 (95.1) | 2.0 (2.0) | 35.4 (35.4) | 0.06 (0.06) | 3.8 (3.8) | 99.93 (99.93) | 98.85 (98.85) |
| E30:Replace | 95.6 (95.4) | 2.5 (2.2) | 688 (689) | 1.25 (1.25) | 1.2 (1.2) | 98.68 (98.68) | 75.18 (75.12) |
| E30:Majority Vote | 95.9 (95.9) | 2.8 (2.8) | 380 (395) | 0.69 (0.72) | 1.4 (1.4) | 99.27 (99.24) | 86.44 (85.83) |
| E30:MB | 96.0 (96.1) | 2.8 (2.9) | 253 (277) | 0.46 (0.5) | 1.6 (1.5) | 99.51 (99.47) | 91.14 (90.39) |
| E30:MBME | 96.0 (96.2) | 2.8 (3.0) | 96 (123) | 0.17 (0.22) | 2.5 (2.2) | 99.82 (99.76) | 96.45 (95.68) |
| E30:CR 2 | 96.0 (96.2) | 2.8 (3.0) | 192 (147) | 0.35 (0.27) | 1.7 (2.0) | 99.63 (99.72) | 93.12 (94.5) |
| E30:CR 5 | 96.0 (96.2) | 2.8 (3.0) | 103 (98) | 0.19 (0.18) | 2.4 (2.6) | 99.8 (99.81) | 96.07 (96.36) |
| E30:CR 10 | 96.0 (96.2) | 2.9 (3.0) | 89 (86) | 0.16 (0.16) | 2.6 (2.7) | 99.83 (99.83) | 96.67 (96.85) |
| R20:Replace | 94.5 (94.5) | 1.3 (1.3) | 274.6 (274.6) | 0.5 (0.5) | 1.2 (1.2) | 99.47 (99.47) | 91.94 (91.94) |
| R20:Majority Vote | 94.9 (94.9) | 1.7 (1.7) | 195.2 (195.2) | 0.35 (0.35) | 1.4 (1.4) | 99.62 (99.62) | 94.06 (94.06) |
| R20:MB | 94.7 (94.8) | 1.6 (1.7) | 107.0 (120.4) | 0.19 (0.22) | 1.7 (1.7) | 99.79 (99.77) | 96.86 (96.43) |
| R20:MBME | 94.6 (94.6) | 1.4 (1.4) | 31.0 (45.4) | 0.06 (0.08) | 3.3 (2.6) | 99.94 (99.91) | 99.09 (98.67) |
| R20:CR 2 | 94.5 (94.5) | 1.4 (1.4) | 65.0 (48.6) | 0.12 (0.09) | 2.0 (2.4) | 99.87 (99.91) | 98.08 (98.55) |
| R20:CR 5 | 94.7 (94.7) | 1.5 (1.5) | 26.6 (27.4) | 0.05 (0.05) | 3.8 (3.7) | 99.95 (99.95) | 99.19 (99.16) |
| R20:CR 10 | 94.6 (94.6) | 1.5 (1.4) | 24.6 (24.4) | 0.04 (0.04) | 4.0 (3.9) | 99.95 (99.95) | 99.25 (99.26) |
| E20:Replace | 95.8 (95.7) | 2.6 (2.5) | 462 (478) | 0.84 (0.87) | 1.3 (1.3) | 99.11 (99.08) | 83.35 (83.11) |
| E20:Majority Vote | 95.9 (96.0) | 2.8 (2.8) | 336 (347) | 0.61 (0.63) | 1.4 (1.4) | 99.35 (99.33) | 88.2 (87.94) |
| E20:MB | 96.0 (95.9) | 2.9 (2.8) | 217 (263) | 0.39 (0.48) | 1.7 (1.5) | 99.58 (99.49) | 92.45 (91.02) |
| E20:MBME | 96.0 (96.0) | 2.9 (2.8) | 82 (115) | 0.15 (0.21) | 2.8 (2.2) | 99.84 (99.78) | 97.02 (96.04) |
| E20:CR 2 | 96.1 (96.9) | 2.9 (2.8) | 156 (130) | 0.28 (0.24) | 1.9 (2.1) | 99.7 (99.75) | 94.53 (95.27) |
| E20:CR 5 | 96.0 (95.9) | 2.9 (2.8) | 84 (90) | 0.15 (0.16) | 2.7 (2.5) | 99.84 (99.83) | 96.86 (96.71) |
| E20:CR 10 | 96.0 (95.9) | 2.9 (2.8) | 75 (80) | 0.14 (0.15) | 2.9 (2.7) | 99.86 (99.85) | 97.22 (97.11) |
| R10:Replace | 94.1 (94.1) | 0.9 (0.9) | 129.2 (129.2) | 0.23 (0.23) | 1.3 (1.3) | 99.75 (99.75) | 96.31 (96.31) |
| R10:Majority Vote | 94.3 (94.3) | 1.1 (1.1) | 109.4 (109.4) | 0.2 (0.2) | 1.5 (1.5) | 99.79 (99.79) | 96.86 (96.86) |
| R10:MB | 94.0 (94.2) | 0.9 (1.0) | 64.0 (70.8) | 0.12 (0.13) | 1.7 (1.7) | 99.88 (99.86) | 98.23 (98.0) |
| R10:MBME | 93.8 (93.8) | 0.6 (0.6) | 14.2 (21.4) | 0.03 (0.04) | 3.3 (2.4) | 99.97 (99.96) | 99.61 (99.41) |
| R10:CR 2 | 93.8 (93.8) | 0.7 (0.7) | 35.2 (23.8) | 0.06 (0.04) | 2.0 (2.4) | 99.93 (99.95) | 99.03 (99.34) |
| R10:CR 5 | 93.7 (93.7) | 0.5 (0.6) | 11.0 (11.0) | 0.02 (0.02) | 3.5 (3.6) | 99.98 (99.98) | 99.69 (99.69) |
| R10:CR 10 | 93.8 (93.8) | 0.7 (0.7) | 6.8 (7.0) | 0.01 (0.01) | 6.0 (5.9) | 99.99 (99.99) | 99.81 (99.8) |
| E10:Replace | 95.3 (95.4) | 2.1 (2.2) | 206 (201) | 0.37 (0.37) | 1.5 (1.5) | 99.6 (99.61) | 93.45 (93.65) |
| E10:Majority Vote | 95.4 (95.4) | 2.2 (2.2) | 200 (200) | 0.36 (0.36) | 1.6 (1.6) | 99.61 (99.61) | 93.67 (93.69) |
| E10:MB | 95.3 (95.3) | 2.1 (2.1) | 140 (145) | 0.25 (0.26) | 1.8 (1.7) | 99.73 (99.72) | 95.59 (95.45) |
| E10:MBME | 95.2 (95.2) | 2.0 (2.0) | 35 (66) | 0.06 (0.12) | 3.9 (2.5) | 99.93 (99.87) | 98.92 (98.02) |
| E10:CR 2 | 95.2 (95.2) | 2.1 (2.1) | 91 (59) | 0.17 (0.11) | 2.1 (2.7) | 99.82 (99.89) | 97.26 (98.11) |
| E10:CR 5 | 95.2 (95.2) | 2.0 (2.0) | 30 (38) | 0.05 (0.07) | 4.3 (3.7) | 99.94 (99.93) | 99.05 (98.79) |
| E10:CR 10 | 95.2 (95.2) | 2.0 (2.0) | 29 (36) | 0.05 (0.07) | 4.4 (3.8) | 99.94 (99.93) | 99.09 (98.86) |

Table 5: Full results of all experiments on **CIFAR-10 (left) and ImageNet (right)** under the standard setting of an **improving sequence of models** and **estimating confusion matrices on a separate split** of the data (i.e., not on the target data set) as explained in § 6.1. The first number in each cell refers to estimating only the diagonal elements (i.e., the class-specific accuracies) of confusion matrices, and the **second number (in brackets) refers to estimating the full confusion matrices** with smoothing **and incorporating soft labels**. The character **E** in front of a budget indicates that selection for re-evaluation is based on our posterior label **entropy criterion** (e.g., E30 refers to selecting the top 30% samples with highest entropy), and the character **R** indicates selecting a **randomly** sampled subset without replacement. Note that entropy-based selection requires a posterior and is thus not applicable for the baselines Replace and Majority Vote.

**CIFAR-10**

| Strategy Budget % | Acc (%) | ΔAcc (%) | Σ NF | NFR (%) | PF / NF | Avg. BTC | Avg. BEC |
|---|---|---|---|---|---|---|---|
| 100:Oracle | 99.3 (99.3) | 6.1 (6.1) | 0 (0) | 0.0 (0.0) | NaN (NaN) | 100.0 (100.0) | 100.0 (100.0) |
| R100:Replace | 94.7 (94.7) | 1.5 (1.5) | 1418 (1418) | 2.58 (2.58) | 1.1 (1.1) | 97.26 (97.26) | 54.79 (54.79) |
| R100:Majority Vote | 96.1 (96.1) | 2.9 (2.9) | 502 (502) | 0.91 (0.91) | 1.3 (1.3) | 99.03 (99.03) | 81.94 (81.94) |
| E100:MB | 95.9 (96.2) | 2.8 (3.0) | 284 (252) | 0.52 (0.46) | 1.5 (1.6) | 99.45 (99.52) | 89.64 (89.69) |
| E100:MBME | 96.1 (96.1) | 2.9 (3.0) | 119 (173) | 0.22 (0.31) | 2.2 (1.9) | 99.77 (99.67) | 95.28 (92.98) |
| E100:CR 2 | 96.0 (96.1) | 2.8 (3.0) | 219 (170) | 0.4 (0.31) | 1.6 (1.9) | 99.58 (99.68) | 91.69 (92.98) |
| E100:CR 5 | 95.9 (96.2) | 2.8 (3.0) | 132 (103) | 0.24 (0.19) | 2.1 (2.5) | 99.75 (99.8) | 94.65 (95.68) |
| E100:CR 10 | 96.0 (96.2) | 2.9 (3.0) | 112 (73) | 0.2 (0.13) | 2.3 (3.1) | 99.79 (99.86) | 95.49 (96.99) |
| R50:Replace | 94.7 (94.7) | 1.6 (1.6) | 711.4 (711.4) | 1.29 (1.29) | 1.1 (1.1) | 98.62 (98.62) | 77.8 (77.8) |
| R50:Majority Vote | 95.7 (95.7) | 2.6 (2.6) | 327.4 (327.4) | 0.6 (0.6) | 1.4 (1.4) | 99.37 (99.37) | 89.06 (89.06) |
| R50:MB | 95.8 (96.0) | 2.7 (2.8) | 196.4 (155.0) | 0.36 (0.28) | 1.7 (1.9) | 99.62 (99.7) | 93.41 (94.27) |
| R50:MBME | 95.7 (95.9) | 2.5 (2.7) | 69.8 (106.0) | 0.13 (0.19) | 2.8 (2.3) | 99.87 (99.8) | 97.58 (96.2) |
| R50:CR 2 | 95.8 (96.0) | 2.6 (2.9) | 133.8 (99.8) | 0.24 (0.18) | 2.0 (2.4) | 99.74 (99.81) | 95.32 (96.25) |
| R50:CR 5 | 95.7 (96.0) | 2.5 (2.8) | 73.2 (59.2) | 0.13 (0.11) | 2.7 (3.4) | 99.86 (99.89) | 97.4 (97.77) |
| R50:CR 10 | 95.7 (95.9) | 2.6 (2.7) | 61.0 (40.8) | 0.11 (0.07) | 3.1 (4.3) | 99.88 (99.92) | 97.83 (98.47) |
| E50:Replace | 95.2 (95.2) | 2.0 (2.0) | 1019 (1012) | 1.85 (1.84) | 1.1 (1.1) | 98.04 (98.05) | 64.97 (64.69) |
| E50:Majority Vote | 96.1 (96.0) | 2.9 (2.9) | 428 (419) | 0.78 (0.76) | 1.3 (1.3) | 99.18 (99.19) | 84.48 (84.84) |
| E50:MB | 96.1 (96.2) | 2.9 (3.0) | 260 (257) | 0.47 (0.47) | 1.6 (1.6) | 99.5 (99.51) | 90.63 (89.36) |
| E50:MBME | 96.1 (96.1) | 3.0 (3.0) | 105 (170) | 0.19 (0.31) | 2.4 (1.9) | 99.8 (99.67) | 96.04 (93.08) |
| E50:CR 2 | 96.0 (96.1) | 2.8 (3.0) | 206 (170) | 0.37 (0.31) | 1.7 (1.9) | 99.6 (99.68) | 92.4 (92.92) |
| E50:CR 5 | 96.0 (96.2) | 2.9 (3.0) | 123 (102) | 0.22 (0.19) | 2.2 (2.5) | 99.76 (99.81) | 95.18 (95.72) |
| E50:CR 10 | 96.1 (96.2) | 3.0 (3.0) | 99 (71) | 0.18 (0.13) | 2.5 (3.1) | 99.81 (99.86) | 96.21 (97.07) |
| R30:Replace | 94.6 (94.6) | 1.4 (1.4) | 439.6 (439.6) | 0.8 (0.8) | 1.2 (1.2) | 99.15 (99.15) | 86.74 (86.74) |
| R30:Majority Vote | 95.3 (95.3) | 2.2 (2.2) | 238.4 (238.4) | 0.43 (0.43) | 1.5 (1.5) | 99.54 (99.54) | 92.54 (92.54) |
| R30:MB | 95.2 (95.6) | 2.1 (2.4) | 146.2 (117.6) | 0.27 (0.21) | 1.7 (2.0) | 99.72 (99.77) | 95.46 (95.92) |
| R30:MBME | 95.2 (95.6) | 2.0 (2.4) | 47.0 (88.4) | 0.09 (0.16) | 3.2 (2.4) | 99.91 (99.83) | 98.47 (96.94) |
| R30:CR 2 | 95.2 (95.5) | 2.0 (2.4) | 88.8 (72.6) | 0.16 (0.13) | 2.1 (2.6) | 99.83 (99.86) | 97.16 (97.5) |
| R30:CR 5 | 95.1 (95.4) | 1.9 (2.3) | 45.6 (40.8) | 0.08 (0.07) | 3.1 (3.8) | 99.91 (99.92) | 98.54 (98.61) |
| R30:CR 10 | 95.2 (95.4) | 2.0 (2.2) | 35.4 (28.8) | 0.06 (0.05) | 3.8 (4.8) | 99.93 (99.94) | 98.85 (99.03) |
| E30:Replace | 95.4 (95.4) | 2.5 (2.2) | 688 (689) | 1.25 (1.25) | 1.2 (1.2) | 98.68 (98.68) | 75.18 (75.12) |
| E30:Majority Vote | 95.9 (95.9) | 2.8 (2.8) | 380 (395) | 0.69 (0.72) | 1.4 (1.4) | 99.27 (99.24) | 86.44 (85.83) |
| E30:MB | 96.0 (96.3) | 2.8 (3.1) | 253 (230) | 0.46 (0.42) | 1.6 (1.7) | 99.53 (99.56) | 91.14 (90.39) |
| E30:MBME | 96.0 (96.2) | 2.8 (3.0) | 96 (163) | 0.17 (0.3) | 2.5 (1.9) | 99.82 (99.69) | 96.45 (93.3) |
| E30:CR 2 | 96.0 (96.2) | 2.8 (3.1) | 192 (160) | 0.35 (0.29) | 1.7 (2.0) | 99.63 (99.69) | 93.12 (93.28) |
| E30:CR 5 | 96.0 (96.2) | 2.8 (3.1) | 103 (94) | 0.19 (0.17) | 2.4 (2.6) | 99.8 (99.82) | 96.07 (96.01) |
| E30:CR 10 | 96.0 (96.3) | 2.9 (3.1) | 89 (57) | 0.16 (0.1) | 2.6 (3.8) | 99.83 (99.89) | 96.67 (97.65) |
| R20:Replace | 94.5 (94.5) | 1.3 (1.3) | 274.6 (274.6) | 0.5 (0.5) | 1.2 (1.2) | 99.47 (99.47) | 91.94 (91.94) |
| R20:Majority Vote | 94.9 (94.9) | 1.7 (1.7) | 195.2 (195.2) | 0.35 (0.35) | 1.4 (1.4) | 99.62 (99.62) | 94.06 (94.06) |
| R20:MB | 94.7 (95.3) | 1.6 (2.2) | 107.0 (83.2) | 0.19 (0.15) | 1.7 (2.3) | 99.79 (99.84) | 96.86 (97.29) |
| R20:MBME | 94.6 (95.3) | 1.4 (2.2) | 31.0 (62.0) | 0.06 (0.11) | 3.3 (2.7) | 99.94 (99.88) | 99.09 (97.98) |
| R20:CR 2 | 94.5 (95.2) | 1.4 (2.0) | 65.0 (56.4) | 0.12 (0.1) | 2.0 (2.8) | 99.87 (99.89) | 98.08 (98.17) |
| R20:CR 5 | 94.7 (95.2) | 1.5 (2.0) | 26.6 (25.6) | 0.05 (0.05) | 3.8 (4.9) | 99.95 (99.95) | 99.19 (99.16) |
| R20:CR 10 | 94.6 (94.9) | 1.5 (1.8) | 24.6 (19.6) | 0.04 (0.04) | 4.0 (5.6) | 99.95 (99.96) | 99.25 (99.38) |
| E20:Replace | 95.8 (95.7) | 2.6 (2.5) | 462 (478) | 0.84 (0.87) | 1.3 (1.3) | 99.11 (99.08) | 83.35 (83.11) |
| E20:Majority Vote | 95.9 (96.0) | 2.8 (2.8) | 336 (347) | 0.61 (0.63) | 1.4 (1.4) | 99.35 (99.33) | 88.2 (87.94) |
| E20:MB | 96.0 (96.0) | 2.9 (2.8) | 217 (207) | 0.39 (0.38) | 1.7 (1.7) | 99.58 (99.6) | 92.45 (91.74) |
| E20:MBME | 96.0 (95.9) | 2.9 (2.8) | 82 (148) | 0.15 (0.27) | 2.8 (1.9) | 99.84 (99.72) | 97.02 (94.17) |
| E20:CR 2 | 96.1 (96.0) | 2.9 (2.8) | 156 (133) | 0.28 (0.24) | 1.9 (2.1) | 99.7 (99.75) | 94.53 (94.67) |
| E20:CR 5 | 96.0 (96.0) | 2.9 (2.9) | 84 (80) | 0.15 (0.15) | 2.7 (2.8) | 99.84 (99.85) | 96.86 (96.73) |
| E20:CR 10 | 96.0 (96.0) | 2.9 (2.9) | 75 (55) | 0.14 (0.1) | 2.9 (3.7) | 99.86 (99.89) | 97.22 (97.89) |
| R10:Replace | 94.1 (94.1) | 0.9 (0.9) | 129.2 (129.2) | 0.23 (0.23) | 1.3 (1.3) | 99.75 (99.75) | 96.31 (96.31) |
| R10:Majority Vote | 94.3 (94.3) | 1.1 (1.1) | 109.4 (109.4) | 0.2 (0.2) | 1.5 (1.5) | 99.79 (99.79) | 96.86 (96.86) |
| R10:MB | 94.0 (94.7) | 0.9 (1.6) | 64.0 (51.0) | 0.12 (0.09) | 1.7 (2.5) | 99.88 (99.9) | 98.23 (98.49) |
| R10:MBME | 93.8 (94.6) | 0.6 (1.4) | 14.2 (36.2) | 0.03 (0.07) | 3.3 (2.9) | 99.97 (99.93) | 99.61 (98.93) |
| R10:CR 2 | 93.8 (94.7) | 0.7 (1.5) | 35.2 (32.4) | 0.06 (0.06) | 2.0 (3.3) | 99.93 (99.94) | 99.03 (99.03) |
| R10:CR 5 | 93.7 (94.3) | 0.5 (1.1) | 11.0 (14.0) | 0.02 (0.03) | 3.5 (5.1) | 99.98 (99.97) | 99.69 (99.59) |
| R10:CR 10 | 93.8 (94.3) | 0.7 (1.1) | 6.8 (7.4) | 0.01 (0.01) | 6.0 (8.5) | 99.99 (99.99) | 99.81 (99.78) |
| E10:Replace | 95.3 (95.4) | 2.1 (2.2) | 206 (201) | 0.37 (0.37) | 1.5 (1.5) | 99.6 (99.61) | 93.45 (93.65) |
| E10:Majority Vote | 95.4 (95.4) | 2.2 (2.2) | 200 (200) | 0.36 (0.36) | 1.6 (1.6) | 99.61 (99.61) | 93.67 (93.69) |
| E10:MB | 95.3 (95.7) | 2.1 (2.5) | 140 (120) | 0.25 (0.22) | 1.8 (2.0) | 99.73 (99.77) | 95.59 (95.48) |
| E10:MBME | 95.2 (95.6) | 2.0 (2.5) | 35 (99) | 0.06 (0.18) | 3.9 (2.3) | 99.93 (99.81) | 98.92 (96.32) |
| E10:CR 2 | 95.2 (95.7) | 2.1 (2.5) | 91 (78) | 0.17 (0.14) | 2.1 (2.6) | 99.82 (99.85) | 97.26 (97.08) |
| E10:CR 5 | 95.2 (95.6) | 2.0 (2.4) | 30 (52) | 0.05 (0.09) | 4.3 (3.3) | 99.94 (99.9) | 99.05 (98.04) |
| E10:CR 10 | 95.2 (95.5) | 2.0 (2.3) | 29 (37) | 0.05 (0.07) | 4.4 (4.1) | 99.94 (99.93) | 99.09 (98.73) |

**ImageNet**

| Strategy (Budget %) | Acc (%) | ΔAcc (%) | Σ NF | NFR (%) | PF / NF | Avg. BTC | Avg. BEC |
|---|---|---|---|---|---|---|---|
| 100:Oracle | 91.2 (91.2) | 34.7 (34.7) | 0 (0) | 0.0 (0.0) | NaN (NaN) | 100.0 (100.0) | 100.0 (100.0) |
| R100:Replace | 79.2 (79.2) | 22.7 (22.7) | 24214 (24214) | 6.05 (6.05) | 1.2 (1.2) | 91.37 (91.37) | 77.71 (77.71) |
| R100:Majority Vote | 78.9 (78.9) | 22.3 (22.3) | 7352 (7352) | 1.84 (1.84) | 1.8 (1.8) | 97.18 (97.18) | 93.95 (93.95) |
| E100:MB | 77.1 (77.6) | 20.5 (21.0) | 4378 (3463) | 1.09 (0.87) | 2.2 (2.5) | 98.32 (98.65) | 96.45 (97.25) |
| E100:MBME | 77.3 (77.4) | 20.7 (20.8) | 3057 (3207) | 0.76 (0.8) | 2.7 (2.6) | 98.78 (98.74) | 97.69 (97.51) |
| E100:CR 2 | 77.1 (77.3) | 20.6 (20.8) | 3368 (1648) | 0.84 (0.41) | 2.5 (4.2) | 98.72 (99.36) | 97.19 (98.7) |
| E100:CR 5 | 77.1 (77.0) | 20.5 (20.5) | 2520 (990) | 0.63 (0.25) | 3.0 (6.2) | 99.06 (99.63) | 97.82 (99.19) |
| E100:CR 10 | 77.0 (76.8) | 20.5 (20.3) | 2112 (788) | 0.53 (0.2) | 3.4 (7.4) | 99.22 (99.71) | 98.15 (99.34) |
| R50:Replace | 78.4 (78.4) | 21.8 (21.8) | 11478.8 (11463.8) | 2.87 (2.87) | 1.5 (1.5) | 95.82 (95.83) | 89.88 (89.89) |
| R50:Majority Vote | 78.1 (78.1) | 21.6 (21.6) | 5048.2 (5048.8) | 1.26 (1.26) | 2.1 (2.1) | 98.07 (98.07) | 95.92 (95.92) |
| R50:MB | 76.9 (76.9) | 20.4 (20.4) | 3022.8 (2509.8) | 0.76 (0.63) | 2.7 (3.0) | 98.83 (99.02) | 97.62 (98.05) |
| R50:MBME | 76.5 (76.7) | 20.0 (20.1) | 2325.4 (2437.2) | 0.58 (0.61) | 3.1 (3.1) | 99.08 (99.05) | 98.27 (98.12) |
| R50:CR 2 | 76.8 (76.5) | 20.3 (19.9) | 2233.6 (1220.8) | 0.56 (0.31) | 3.3 (5.1) | 99.14 (99.55) | 98.22 (99.06) |
| R50:CR 5 | 76.6 (76.1) | 20.1 (19.5) | 1603.8 (700.4) | 0.4 (0.18) | 4.1 (8.0) | 99.39 (99.73) | 98.69 (99.44) |
| R50:CR 10 | 76.7 (75.8) | 20.1 (19.2) | 1314.6 (530.0) | 0.33 (0.13) | 4.8 (10.1) | 99.51 (99.8) | 98.92 (99.57) |
| E50:Replace | 78.9 (79.8) | 22.4 (23.2) | 15732 (16138) | 3.93 (4.03) | 1.4 (1.4) | 94.41 (94.3) | 85.37 (84.73) |
| E50:Majority Vote | 78.7 (79.2) | 22.2 (22.6) | 6318 (6282) | 1.58 (1.57) | 1.9 (1.9) | 97.6 (97.64) | 94.74 (94.67) |
| E50:MB | 77.8 (77.9) | 21.3 (21.4) | 3969 (3212) | 0.99 (0.8) | 2.3 (2.7) | 98.48 (98.77) | 96.78 (97.4) |
| E50:MBME | 77.6 (77.6) | 21.1 (21.1) | 2904 (2972) | 0.73 (0.74) | 2.8 (2.8) | 98.85 (98.85) | 97.79 (97.66) |
| E50:CR 2 | 77.8 (77.7) | 21.2 (21.1) | 3014 (1547) | 0.75 (0.39) | 2.8 (4.4) | 98.86 (99.41) | 97.5 (98.75) |
| E50:CR 5 | 77.7 (77.3) | 21.2 (20.8) | 2214 (900) | 0.55 (0.22) | 3.4 (6.8) | 99.08 (99.24) | 98.09 (99.24) |
| E50:CR 10 | 77.7 (77.1) | 21.2 (20.6) | 1832 (709) | 0.46 (0.18) | 3.9 (8.3) | 99.33 (99.74) | 98.4 (99.38) |
| R30:Replace | 77.4 (77.3) | 20.8 (20.8) | 6546.4 (6537.8) | 1.64 (1.63) | 1.8 (1.8) | 97.56 (97.56) | 94.53 (94.55) |
| R30:Majority Vote | 77.1 (77.1) | 20.5 (20.5) | 3616.4 (3613.6) | 0.9 (0.9) | 2.4 (2.4) | 98.6 (98.6) | 97.18 (97.18) |
| R30:MB | 75.9 (75.9) | 19.4 (19.4) | 2186.2 (1860.6) | 0.55 (0.47) | 3.2 (3.6) | 99.14 (99.27) | 98.35 (98.6) |
| R30:MBME | 75.3 (75.8) | 18.7 (19.3) | 1859.6 (1855.6) | 0.46 (0.46) | 3.5 (3.6) | 99.26 (99.27) | 98.65 (98.61) |
| R30:CR 2 | 75.5 (75.3) | 19.0 (18.7) | 1607.0 (918.2) | 0.4 (0.23) | 4.0 (6.1) | 99.37 (99.64) | 98.8 (99.31) |
| R30:CR 5 | 75.1 (74.5) | 18.5 (17.9) | 1087.2 (488.4) | 0.27 (0.12) | 5.3 (10.2) | 99.58 (99.81) | 99.17 (99.63) |
| R30:CR 10 | 74.9 (73.9) | 18.4 (17.3) | 875.4 (341.8) | 0.22 (0.09) | 6.2 (13.7) | 99.66 (99.87) | 99.33 (99.74) |
| E30:Replace | 78.5 (79.1) | 22.0 (22.6) | 9708 (8764) | 2.43 (2.19) | 1.6 (1.6) | 96.53 (96.87) | 91.01 (91.87) |
| E30:Majority Vote | 78.5 (78.9) | 22.0 (22.3) | 5232 (4469) | 1.31 (1.12) | 2.1 (2.2) | 98.03 (98.32) | 95.63 (96.24) |
| E30:MB | 78.1 (78.1) | 21.6 (21.5) | 3375 (2450) | 0.84 (0.61) | 2.6 (3.2) | 98.71 (99.06) | 97.25 (98.02) |
| E30:MBME | 77.8 (77.8) | 21.2 (21.3) | 2577 (2407) | 0.64 (0.6) | 3.1 (3.2) | 98.98 (99.07) | 98.04 (98.07) |
| E30:CR 2 | 78.0 (77.7) | 21.5 (21.1) | 2578 (1221) | 0.64 (0.31) | 3.1 (5.3) | 99.02 (99.53) | 97.86 (99.02) |
| E30:CR 5 | 78.0 (77.4) | 21.5 (20.8) | 1831 (670) | 0.46 (0.17) | 3.9 (8.8) | 99.32 (99.75) | 98.43 (99.43) |
| E30:CR 10 | 77.9 (77.1) | 21.4 (20.6) | 1517 (491) | 0.38 (0.12) | 4.5 (11.5) | 99.44 (99.82) | 98.67 (99.57) |
| R20:Replace | 75.7 (75.7) | 19.1 (19.1) | 4171.0 (4183.8) | 1.04 (1.05) | 2.1 (2.1) | 98.41 (98.41) | 96.71 (96.7) |
| R20:Majority Vote | 75.4 (75.4) | 18.9 (18.8) | 2690.2 (2696.6) | 0.67 (0.67) | 2.8 (2.7) | 98.95 (98.94) | 97.99 (97.98) |
| R20:MB | 74.0 (74.6) | 17.5 (18.0) | 1662.8 (1421.2) | 0.42 (0.36) | 3.6 (4.2) | 99.34 (99.44) | 98.81 (98.97) |
| R20:MBME | 73.5 (74.3) | 16.9 (17.8) | 1480.6 (1421.8) | 0.37 (0.36) | 3.9 (4.1) | 99.4 (99.43) | 98.97 (98.97) |
| R20:CR 2 | 73.2 (73.6) | 16.7 (17.0) | 1205.6 (718.2) | 0.3 (0.18) | 4.5 (6.9) | 99.52 (99.71) | 99.15 (99.48) |
| R20:CR 5 | 72.3 (72.2) | 15.8 (15.7) | 768.0 (344.0) | 0.19 (0.09) | 6.1 (12.4) | 99.69 (99.86) | 99.46 (99.75) |
| R20:CR 10 | 71.9 (71.0) | 15.4 (14.4) | 603.8 (218.4) | 0.15 (0.05) | 7.4 (17.5) | 99.76 (99.91) | 99.57 (99.84) |
| E20:Replace | 78.5 (78.4) | 22.0 (21.8) | 6191 (5239) | 1.55 (1.31) | 1.9 (2.0) | 97.74 (98.1) | 94.47 (95.3) |
| E20:Majority Vote | 78.3 (78.0) | 21.8 (21.4) | 4295 (3258) | 1.07 (0.81) | 2.3 (2.6) | 98.37 (98.77) | 96.46 (97.32) |
| E20:MB | 77.9 (77.2) | 21.3 (20.6) | 2700 (1825) | 0.68 (0.46) | 3.0 (3.8) | 98.96 (99.29) | 97.86 (98.58) |
| E20:MBME | 77.6 (77.2) | 21.1 (20.6) | 2183 (1825) | 0.55 (0.46) | 3.4 (3.8) | 99.13 (99.29) | 98.39 (98.58) |
| E20:CR 2 | 77.8 (76.6) | 21.3 (20.1) | 1999 (879) | 0.5 (0.22) | 3.7 (6.7) | 99.23 (99.66) | 98.41 (99.32) |
| E20:CR 5 | 77.7 (75.9) | 21.2 (19.4) | 1383 (448) | 0.35 (0.11) | 4.8 (11.8) | 99.47 (99.83) | 98.87 (99.63) |
| E20:CR 10 | 77.7 (75.1) | 21.2 (18.6) | 1101 (266) | 0.28 (0.07) | 5.8 (18.4) | 99.58 (99.9) | 99.08 (99.78) |
| R10:Replace | 71.3 (71.3) | 14.7 (14.8) | 1958.4 (1952.6) | 0.49 (0.49) | 2.9 (2.9) | 99.22 (99.22) | 98.63 (98.63) |
| R10:Majority Vote | 71.2 (71.2) | 14.7 (14.6) | 1481.4 (1489.6) | 0.37 (0.37) | 3.5 (3.5) | 99.4 (99.4) | 98.98 (98.98) |
| R10:MB | 69.0 (70.0) | 12.5 (13.4) | 991.4 (865.4) | 0.25 (0.22) | 4.1 (4.9) | 99.59 (99.65) | 99.35 (99.42) |
| R10:MBME | 68.9 (70.1) | 12.4 (13.6) | 944.8 (878.4) | 0.24 (0.22) | 4.3 (4.9) | 99.61 (99.64) | 99.38 (99.41) |
| R10:CR 2 | 67.9 (69.2) | 11.4 (12.6) | 703.6 (424.2) | 0.18 (0.11) | 5.0 (8.4) | 99.71 (99.83) | 99.54 (99.72) |
| R10:CR 5 | 66.2 (66.9) | 9.6 (10.4) | 393.4 (161.8) | 0.1 (0.04) | 7.1 (17.1) | 99.84 (99.93) | 99.75 (99.89) |
| R10:CR 10 | 65.3 (64.9) | 8.8 (8.3) | 282.6 (89.0) | 0.07 (0.02) | 8.7 (24.4) | 99.88 (99.96) | 99.82 (99.94) |
| E10:Replace | 76.1 (77.0) | 19.5 (20.4) | 2468 (2216) | 0.62 (0.55) | 3.0 (3.3) | 99.04 (99.17) | 98.12 (98.18) |
| E10:Majority Vote | 75.9 (76.1) | 19.3 (19.5) | 2417 (1954) | 0.6 (0.49) | 3.0 (3.5) | 99.06 (99.25) | 98.18 (98.45) |
| E10:MB | 75.3 (75.2) | 18.8 (18.7) | 1557 (1118) | 0.39 (0.28) | 4.0 (5.2) | 99.38 (99.56) | 98.89 (99.18) |
| E10:MBME | 75.2 (75.2) | 18.7 (18.7) | 1533 (1118) | 0.38 (0.28) | 4.0 (5.2) | 99.38 (99.56) | 98.87 (99.18) |
| E10:CR 2 | 75.3 (74.3) | 18.7 (17.8) | 1118 (503) | 0.28 (0.13) | 5.2 (9.8) | 99.55 (99.8) | 99.22 (99.63) |
| E10:CR 5 | 75.2 (72.7) | 18.6 (16.2) | 700 (165) | 0.18 (0.04) | 7.7 (25.5) | 99.72 (99.93) | 99.51 (99.88) |
| E10:CR 10 | 75.2 (70.7) | 18.6 (14.2) | 515 (71) | 0.13 (0.02) | 10.1 (50.8) | 99.79 (99.97) | 99.64 (99.95) |

Table 6: Full results of all experiments on **ObjectNet** under the standard setting of an **improving sequence of models** and **estimating confusion matrices on a separate split** of the data (i.e., not on the target data set) as explained in § 6.1. The first number in each cell refers to estimating only the diagonal elements (i.e., the class-specific accuracies) of confusion matrices, and the **second number (in brackets) refers to estimating the full confusion matrices** with smoothing **and incorporating soft labels**. The character **E** in front of a budget indicates that selection for re-evaluation is based on our posterior label **entropy criterion** (e.g., E30 refers to selecting the top 30% samples with highest entropy), and the character **R** indicates selecting a **randomly** sampled subset without replacement. Note that entropy-based selection requires a posterior and is thus not applicable for the baselines Replace and Majority Vote.

| Strategy (Budget %) | Acc (%) | ΔAcc (%) | Σ NF | NFR (%) | PF / NF | Avg. BTC | Avg. BEC |
|---|---|---|---|---|---|---|---|
| 100:Oracle | 50.5 (50.5) | 42.6 (42.6) | 0 (0) | 0.0 (0.0) | NaN (NaN) | 100.0 (100.0) | 99.99 (99.99) |
| R100:Replace | 31.9 (31.9) | 24.0 (24.0) | 16669 (16669) | 5.62 (5.62) | 1.3 (1.3) | 72.65 (72.65) | 92.61 (92.61) |
| R100:Majority Vote | 29.6 (29.6) | 21.6 (21.6) | 4690 (4690) | 1.58 (1.58) | 1.9 (1.9) | 89.99 (89.99) | 98.02 (98.02) |
| E100:MB | 29.1 (29.6) | 21.2 (21.7) | 2477 (1984) | 0.83 (0.67) | 2.6 (3.0) | 94.46 (95.55) | 98.96 (99.16) |
| E100:MBME | 28.6 (29.1) | 20.6 (21.1) | 1599 (1774) | 0.54 (0.6) | 3.4 (3.2) | 95.86 (95.83) | 99.34 (99.26) |
| E100:CR 2 | 29.0 (29.2) | 21.0 (21.2) | 1876 (931) | 0.63 (0.31) | 3.1 (5.2) | 95.92 (97.9) | 99.21 (99.61) |
| E100:CR 5 | 28.8 (28.7) | 20.8 (20.7) | 1372 (543) | 0.46 (0.18) | 3.8 (8.1) | 97.18 (98.86) | 99.41 (99.77) |
| E100:CR 10 | 28.7 (28.4) | 20.8 (20.5) | 1084 (422) | 0.37 (0.14) | 4.6 (10.0) | 97.82 (99.16) | 99.54 (99.82) |
| R50:Replace | 30.7 (30.7) | 22.7 (22.7) | 7583.8 (7588.8) | 2.56 (2.56) | 1.6 (1.6) | 86.63 (86.61) | 96.69 (96.69) |
| R50:Majority Vote | 28.6 (28.7) | 20.7 (20.7) | 3281.6 (3291.2) | 1.11 (1.11) | 2.2 (2.2) | 93.01 (93.01) | 98.62 (98.62) |
| R50:MB | 27.8 (28.7) | 19.9 (20.7) | 1676.6 (1410.0) | 0.56 (0.48) | 3.2 (3.7) | 96.13 (96.83) | 99.3 (99.41) |
| R50:MBME | 26.9 (28.3) | 18.9 (20.3) | 1280.4 (1360.8) | 0.43 (0.46) | 3.7 (3.8) | 96.77 (96.89) | 99.48 (99.43) |
| R50:CR 2 | 27.5 (28.1) | 19.6 (20.2) | 1165.4 (663.6) | 0.39 (0.22) | 4.1 (6.6) | 97.3 (98.51) | 99.52 (99.72) |
| R50:CR 5 | 27.2 (27.3) | 19.2 (19.4) | 780.2 (361.4) | 0.26 (0.12) | 5.6 (10.9) | 98.26 (99.23) | 99.67 (99.85) |
| R50:CR 10 | 27.0 (26.7) | 19.1 (18.8) | 599.8 (253.0) | 0.2 (0.09) | 6.9 (14.8) | 98.7 (99.47) | 99.75 (99.89) |
| E50:Replace | 31.0 (31.4) | 23.1 (23.4) | 7882 (7329) | 2.66 (2.47) | 1.5 (1.6) | 86.26 (87.93) | 96.53 (96.75) |
| E50:Majority Vote | 29.5 (29.4) | 21.5 (21.5) | 3895 (3397) | 1.31 (1.14) | 2.0 (2.2) | 91.73 (93.32) | 98.36 (98.55) |
| E50:MB | 28.9 (30.0) | 20.9 (22.1) | 2079 (1610) | 0.7 (0.54) | 2.9 (3.5) | 95.23 (96.66) | 99.13 (99.31) |
| E50:MBME | 28.2 (29.5) | 20.3 (21.5) | 1452 (1534) | 0.49 (0.52) | 3.6 (3.6) | 96.18 (96.75) | 99.41 (99.35) |
| E50:CR 2 | 28.7 (29.6) | 20.8 (21.7) | 1506 (763) | 0.51 (0.26) | 3.6 (6.3) | 96.64 (98.41) | 99.37 (99.67) |
| E50:CR 5 | 28.5 (29.0) | 20.6 (21.1) | 1049 (414) | 0.35 (0.14) | 4.6 (10.5) | 97.83 (99.21) | 99.55 (99.82) |
| E50:CR 10 | 28.4 (28.7) | 20.5 (20.7) | 828 (309) | 0.28 (0.1) | 5.6 (13.4) | 98.35 (99.43) | 99.64 (99.86) |
| R30:Replace | 29.0 (29.0) | 21.0 (21.1) | 4070.6 (4074.0) | 1.37 (1.37) | 2.0 (2.0) | 92.19 (92.21) | 98.26 (98.26) |
| R30:Majority Vote | 27.3 (27.3) | 19.4 (19.4) | 2346.6 (2348.8) | 0.79 (0.79) | 2.5 (2.5) | 94.91 (94.88) | 99.02 (99.02) |
| R30:MB | 25.9 (27.3) | 18.0 (19.4) | 1185.8 (1024.8) | 0.4 (0.35) | 3.8 (4.5) | 97.12 (97.58) | 99.52 (99.58) |
| R30:MBME | 25.1 (27.1) | 17.2 (19.2) | 1008.4 (988.6) | 0.34 (0.33) | 4.2 (4.6) | 97.41 (97.67) | 99.59 (99.59) |
| R30:CR 2 | 25.4 (26.5) | 17.5 (18.5) | 782.2 (482.8) | 0.26 (0.16) | 5.2 (8.1) | 98.05 (98.87) | 99.68 (99.8) |
| R30:CR 5 | 24.6 (25.2) | 16.6 (17.2) | 476.2 (236.2) | 0.16 (0.08) | 7.5 (14.6) | 98.8 (99.45) | 99.81 (99.9) |
| R30:CR 10 | 24.4 (24.5) | 16.4 (16.5) | 331.6 (154.2) | 0.11 (0.05) | 10.2 (20.9) | 99.17 (99.65) | 99.86 (99.93) |
| E30:Replace | 29.0 (30.0) | 21.0 (22.1) | 4316 (3761) | 1.45 (1.27) | 1.9 (2.1) | 91.75 (93.51) | 98.14 (98.35) |
| E30:Majority Vote | 28.2 (28.2) | 20.3 (20.2) | 2970 (2359) | 1.0 (0.79) | 2.3 (2.6) | 93.54 (95.37) | 98.76 (99.00) |
| E30:MB | 27.8 (28.9) | 19.9 (20.9) | 1565 (1015) | 0.53 (0.34) | 3.4 (4.8) | 96.24 (97.78) | 99.35 (99.57) |
| E30:MBME | 26.9 (28.9) | 18.9 (20.9) | 1280 (1015) | 0.43 (0.34) | 3.7 (4.8) | 96.64 (97.78) | 99.48 (99.57) |
| E30:CR 2 | 27.7 (28.1) | 19.7 (20.1) | 1074 (496) | 0.36 (0.17) | 4.4 (8.5) | 97.42 (98.9) | 99.55 (99.79) |
| E30:CR 5 | 27.4 (26.9) | 19.4 (19.0) | 689 (230) | 0.23 (0.08) | 6.2 (16.3) | 98.43 (99.52) | 99.71 (99.9) |
| E30:CR 10 | 27.1 (25.9) | 19.2 (18.0) | 504 (140) | 0.17 (0.05) | 8.1 (24.8) | 98.91 (99.71) | 99.79 (99.94) |
| R20:Replace | 27.0 (27.1) | 19.1 (19.1) | 2453.8 (2436.4) | 0.83 (0.82) | 2.4 (2.5) | 94.82 (94.84) | 98.97 (98.98) |
| R20:Majority Vote | 25.8 (25.8) | 17.8 (17.9) | 1658.8 (1663.4) | 0.56 (0.56) | 3.0 (3.0) | 96.17 (96.16) | 99.32 (99.32) |
| R20:MB | 23.8 (25.3) | 15.8 (17.3) | 887.2 (744.0) | 0.3 (0.25) | 4.3 (5.3) | 97.74 (98.15) | 99.64 (99.7) |
| R20:MBME | 23.5 (25.5) | 15.6 (17.5) | 793.8 (736.0) | 0.27 (0.25) | 4.6 (5.4) | 97.93 (98.18) | 99.68 (99.7) |
| R20:CR 2 | 22.6 (24.3) | 14.7 (16.4) | 530.0 (339.0) | 0.18 (0.11) | 6.2 (10.0) | 98.57 (99.14) | 99.79 (99.86) |
| R20:CR 5 | 21.5 (22.7) | 13.5 (14.8) | 304.2 (160.8) | 0.1 (0.05) | 9.3 (18.1) | 99.14 (99.6) | 99.88 (99.93) |
| R20:CR 10 | 20.9 (21.5) | 12.9 (13.6) | 215.6 (98.8) | 0.07 (0.03) | 12.1 (26.6) | 99.4 (99.74) | 99.91 (99.96) |
| E20:Replace | 26.9 (29.3) | 19.0 (21.3) | 2588 (2149) | 0.87 (0.72) | 2.4 (2.8) | 94.54 (96.18) | 98.91 (99.07) |
| E20:Majority Vote | 26.2 (27.5) | 18.3 (19.6) | 2390 (1742) | 0.81 (0.59) | 2.4 (3.1) | 94.8 (96.61) | 99.0 (99.26) |
| E20:MB | 26.0 (27.7) | 18.1 (19.7) | 1169 (763) | 0.39 (0.26) | 3.9 (5.8) | 97.01 (98.26) | 99.53 (99.68) |
| E20:MBME | 25.8 (27.7) | 17.8 (19.7) | 1131 (763) | 0.38 (0.26) | 3.9 (5.8) | 97.06 (98.26) | 99.54 (99.68) |
| E20:CR 2 | 26.0 (26.6) | 18.1 (18.6) | 670 (353) | 0.23 (0.12) | 6.0 (10.8) | 98.14 (99.16) | 99.73 (99.85) |
| E20:CR 5 | 26.0 (25.1) | 18.0 (17.2) | 369 (132) | 0.12 (0.04) | 10.1 (25.2) | 98.97 (99.7) | 99.85 (99.94) |
| E20:CR 10 | 25.9 (24.2) | 17.9 (16.2) | 248 (73) | 0.08 (0.02) | 14.4 (42.2) | 99.34 (99.83) | 99.9 (99.97) |
| R10:Replace | 22.1 (22.1) | 14.2 (14.2) | 996.6 (997.2) | 0.34 (0.34) | 3.6 (3.6) | 97.49 (97.5) | 99.6 (99.6) |
| R10:Majority Vote | 21.6 (21.5) | 13.6 (13.5) | 808.8 (820.4) | 0.27 (0.28) | 4.1 (4.1) | 97.86 (97.83) | 99.68 (99.67) |
| R10:MB | 19.5 (21.0) | 11.6 (13.1) | 450.2 (371.2) | 0.15 (0.13) | 5.8 (7.6) | 98.7 (98.95) | 99.82 (99.85) |
| R10:MBME | 19.1 (20.8) | 11.1 (12.8) | 446.4 (381.6) | 0.15 (0.13) | 5.6 (7.2) | 98.69 (98.92) | 99.83 (99.85) |
| R10:CR 2 | 17.7 (19.8) | 9.8 (11.9) | 287.2 (177.2) | 0.1 (0.06) | 7.3 (13.4) | 99.11 (99.48) | 99.89 (99.93) |
| R10:CR 5 | 15.6 (17.6) | 7.7 (9.7) | 151.6 (75.4) | 0.05 (0.03) | 10.4 (24.9) | 99.5 (99.77) | 99.94 (99.97) |
| R10:CR 10 | 14.6 (15.8) | 6.7 (7.9) | 94.6 (36.0) | 0.03 (0.01) | 14.1 (41.8) | 99.68 (99.89) | 99.96 (99.99) |
| E10:Replace | 23.7 (25.8) | 15.7 (17.9) | 996 (683) | 0.34 (0.23) | 3.9 (5.9) | 97.5 (98.46) | 99.6 (99.72) |
| E10:Majority Vote | 23.7 (25.4) | 15.7 (17.5) | 996 (663) | 0.34 (0.22) | 3.9 (5.9) | 97.5 (98.48) | 99.6 (99.72) |
| E10:MB | 22.7 (24.4) | 14.8 (16.5) | 696 (364) | 0.23 (0.12) | 4.9 (9.4) | 98.08 (99.03) | 99.72 (99.85) |
| E10:MBME | 22.7 (24.4) | 14.8 (16.5) | 696 (364) | 0.23 (0.12) | 4.9 (9.4) | 98.08 (99.03) | 99.72 (99.85) |
| E10:CR 2 | 20.7 (22.7) | 12.8 (14.8) | 427 (176) | 0.14 (0.06) | 6.6 (16.6) | 98.68 (99.52) | 99.83 (99.93) |
| E10:CR 5 | 18.2 (19.4) | 10.3 (11.4) | 197 (50) | 0.07 (0.02) | 10.7 (43.4) | 99.32 (99.86) | 99.92 (99.98) |
| E10:CR 10 | 17.2 (15.9) | 9.2 (7.9) | 122 (28) | 0.04 (0.01) | 15.0 (53.5) | 99.57 (99.92) | 99.95 (99.99) |

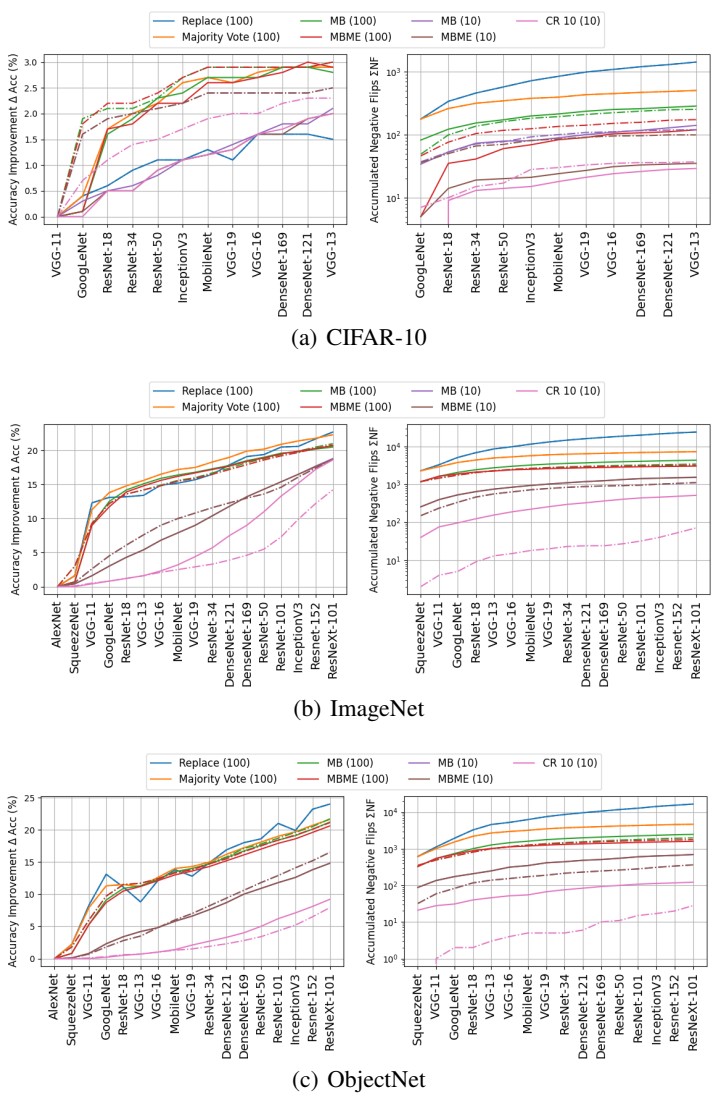

Figure 12: Comparative temporal evolution plots for experiments incorporating uncalibrated softmax labels. Solid lines represent using hard labels with only diagonal estimated for confusion matrix and dashed lines reflect results using soft labels and estimating full confusion matrix with Laplace smoothing.