# OpenReview forum: "Backward-Compatible Prediction Updates: A Probabilistic Approach"
_NeurIPS.cc/2021/Conference — NeurIPS 2021 Poster_

### Official Review · Reviewer_BRHA · 2021-07-01

**Rating:** 7
**Confidence:** 5

**Summary:**

This paper formulates the “Prediction Update Problem”: as new (better) models are trained, you want to integrate them into your system, and improve accuracy. However, you want to do so without making too many previously correct predictions incorrect (negative flips), as this would reduce user trust. The authors provide a natural Bayesian procedure for doing this. Additionally, they consider the case where the dataset may be too large to evaluate each model on the full dataset, and propose an entropy-based subset selection procedure. Experimentally, they show that this method can beat simple baselines at keeping negative flips low without losing too much in accuracy.

**Limitations And Societal Impact:**

Good discussion of the limitations of a number of their assumptions – Section 4 is very nice. I think this work could be useful across industries, in particular high stakes situations and human trust in AI systems

**Main Review:**

-	This problem seems very interesting and well motivated, and the authors describe it well
-	The proposed methods section provides a reasonable working-through of what a sensible, simple approach to this problem might be
-	There are a few assumptions that seem limiting (e.g. independent predictions given label, reliable estimation of confusion matrix) but the authors mostly do a good job mentioning these
-	The experiments tell a clear story although the presentation is quite cramped, it’s hard to really make out what is happening in the figures and charts. But the improvement over the baselines is clear
-	One thing that would be helpful in the experiments is more discussion of the Pareto front between accuracy and negative flips
-	The “limited evaluation budget” part of the story seems a little bit separate – in some sense I’m not sure you really need it as part of the proposed problem
-	BTC + BEC should definitely be defined in the main body!!!
-	The experiment with the models ordered by when the broke SOTA is very cool
-	One baseline I would find interesting would be using the oracle only for estimation of the confusion matrix – that seems like an important source of noise/uncertainty here
-	I’m not sure why the accuracies for the proposed methods decrease with higher budgets? There’s an explanation in the appendix but I don’t really understand it, would love more clarity here
-	Why are the numbers in the \sum NF column floats? Is this an average over seeds?
-	In general, lots of experimental details are left out, there should be a section in the appendix  - I doubt I would be able to reproduce this result as is given the information in the paper currently
-	Sec 6.4: when you discuss the ablations, should point to the appendix where you do the experiment
-	I have no idea what the “reducing re-evaluations matters at scale” section means – I thought you were using VGG/Resnets for your method??
-	Overall: the paper is a little sloppy but very interesting problem proposal and useful experimental evidence


**Time Spent Reviewing:**

2.5

---

> ### Author Response · Authors · 2021-08-10
> **Author response to BRHA**
>
> We thank the reviewer for their useful comments and suggestions. We will use them to improve the current version and are happy to see they find our work and the introduced problem setting very interesting and well motivated.
>
>
> **Float numbers in $\sum$ NF column.**
>
> For all experiments involving the random selection strategies we run the same experiment for five different random seeds each and report the average. Thanks for raising that we missed specifying this detail. We will clarify this in the paper.
>
>
> **Improving upon presentation.**
>
> * We will point in each ablation section to the specific appendix section where we discuss these results in more detail.
> * We will add the formal definition of BTC and BEC to the main body.
> * Figure sizes and captions will be enlarged to improve readability.
> * The code will be released to reproduce all experiments. Also we will make sure to list all experimental details in the appendix, however, we believe the list is almost complete, but we missed to mention that 5 random seeds are used for the random selection ablation studies.
>
>
> **Limited evaluation budget & "Reducing re-evaluations matters at scale.”**
>
> The possibility to reduce the number of evaluations comes naturally from the probabilistic treatment. It also appears in practice in the large-scale scenario and is further influenced by the frequency of model updates. Also consider running distributed computation on edge devices. Therefore we included this relevant dimension, the results for the 100% compute budget are the no-compute-limit results.
>
> Regarding the comment on the “Reducing re-evaluations matters at scale” section. You rightly assume we use VGGs, ResNets in all our experiments. The point we wanted to make is that the re-evaluation of an image (using e.g. biggest ResNet architectures) takes about 550 times longer than a model update step (i.e. computing the approximate posterior, label entropy and update strategy details). This means that re-evaluation dominates compute, justifying the goal of avoiding new evaluations if possible (desiderata 3).
>
>
> **Why the accuracies for the proposed methods decrease with higher budgets?**
>
> We want to clarify that we found this behaviour only in the case of the ImageNet experiments, with a longer discussion including ablation experiments provided in section A.4 in the appendix. Note that for the results reported in 6.2 only the diagonal elements in the confusion matrix (= per-class accuracies) have been estimated. This is possible also for small labelled datasets and so less restrictive in practice. The ablation experiments in A.2 on different confusion matrix estimators, show that this peaking phenomenon is already less pronounced when we would be able to reliably estimate the full confusion matrix from the separate held-out set. We therefore conjecture that this behaviour may be related to inaccurate estimates of the posterior resulting from our approximation of the unknown confusion matrices. To further investigate this hypothesis, we performed an ablation where we use the full validation set both to estimate the confusion matrix (with smoothing) and for evaluation—i.e., we do not use a split of the validation set as in Fig. 2—thus matching almost exactly the confusion counts statistics of the evaluation set (i.e. having access to an oracle estimate of the confusion matrix on this particular evaluation set). We find that the peaking phenomenon disappears almost entirely, with only a very slight drop in accuracy remaining between budgets of 50% and 100% for some strategies. This appears to confirm our hypothesis that the main source of the peaking behaviour are approximation errors in our posterior estimates that are unfortunately unavoidable in practice.
>
>
> **Baseline suggestion: Using the oracle only for estimation of the confusion matrix.**
>
> There may be a misunderstanding. The oracle refers to the no-negative-flips upper limit of performance increase. It receives the model predictions but does not change already correct labels. Estimation of the confusion matrix parameters is always performed on a labelled held-out set. To be able to use pre-trained models, we split the validation set to obtain a separate held-out set for estimation and the rest for evaluation. For ObjectNet we can use the entire ImageNet-val. In case we misunderstand the comment, we kindly ask to expand on this suggestion.
>
>
> **On the Pareto Front between accuracy and negative flips.**
>
> We are happy to expand some discussion of the results in the context of Pareto optimality for the final version.
>
>
> **BTC and BEC.**
>
> We will add the formal definitions to the paper and follow the definition proposed in Srivastava et al. [45] with $h_1$ and $h_2$ being the predictors at time step 1 and time step 2. Translated to our setup, we associate $h_1$ with our estimated and stored prediction at time step 1 whereas $h_2$ corresponds to the updated prediction dataset at time step 2:
>
> **Backward Trust Compatibility (*_BTC_*) score:** *“The ratio of points in a held-out test set (e.g., $D_{\text{test}}$) that $h_2$ predicted correctly among all points $h_1$ had already predicted correctly.”* [45]
>
> $$
> \text{BTC} = \frac{\sum_{i=1}^{|D|} \mathbf{1}[h_1(x_i) = y_i, h_2(x_i) = y_i]}{\sum_{i=1}^{|D|} \mathbf{1}[h_1(x_i) = y_i]}
> $$
>
> **Backward Error Compatibility (*_BEC_*) score:** *“[T]he proportion of points in a held-out test set that $h_2$ predicted incorrectly, out of which $h_1$ also predicted incorrectly, thus capturing the probability that a mistake made by $h_2$ is not new.”* [45]
> $$
> \text{BEC} = \frac{\sum_{i=1}^{|D|} \mathbf{1}[h_1(x_i) \neq y_i, h_2(x_i) \neq y_i]}{\sum_{i=1}^{|D|} \mathbf{1}[h_2(x_i) \neq y_i]}
> $$

---

> > ### Comment · Reviewer_BRHA · 2021-08-13
> > **Response**
> >
> > Thanks for your response.
> >
> > "however, we believe the list is almost complete, but we missed to mention that 5 random seeds are used for the random selection ablation studies."
> >
> > Re: experimental details - I couldn't find a list of hyperparameters used to train the models. If it's in the paper let me know where, if not it should be included in the paper/supplementary somewhere.

---

> > > ### Author Response · Authors · 2021-08-17
> > > **Hyper-parameters of trained models**
> > >
> > > Thank you very much for your reply. We think we now understood the origin of this question:
> > >
> > > **.. couldn’t find a list of hyperparameters used to train the models. If it’s in the paper let me know where**
> > >
> > > We did not list the hyper-parameters used for training the classifiers since we do not consider those being parameters of our model. The problem we consider is to use any given pretrained model **without** knowledge about it has been trained. Everything needed to reproduce our experiments is downloading the fixed models. This is stated and referred to in the main paper (l.273: “For ease of reproducibility, we use pre-trained models from the torchvision model zoo [32] and [34].“) and again in the appendix with links to download in the footnote on p.3 (l.60-l62: “we used models which had been pre-trained on ImageNet, for our experiments on CIFAR-10, we use models which have instead been pre-trained on CIFAR-10, available from an open source github repository.“).
> > >
> > > This “separation of concerns” -- how to train a model and how to update the prediction -- was important to us, therfore we consider it a strength to not require any knowledge about the training procedure of the individual classifiers. We do not retrain or train any models ourselves. Therefore, when it comes to those hyper-parameters we have to refer to the implementations that resulted in the publicly available models. Let us emphasize once more that knowing how these models have been trained is neither needed for the method nor required for reproducing the results.
> > >
> > > We hope this resolves this question and if there is anything else you want us to look into please do not hesitate to let us know. When it comes to our hyper-parameters we are also preparing a code release that should complement the description of the manuscript.

---

> > > > ### Comment · Reviewer_BRHA · 2021-08-17
> > > > **Thanks for the clarification**
> > > >
> > > > Ah my mistake, I forgot about that - thank you!

---

### Official Review · Reviewer_y17T · 2021-07-18

**Rating:** 8
**Confidence:** 4

**Summary:**

Paper proposes a new problem *Prediction Update Problem* with a complementary perspective originating from the increasing availability of pre-trained and regularly improving state-of-the-art models. In particular, they focus on (i) given a limited budget, deciding which data points should be re-evaluated using the new model; and (ii) if the new predictions differ from the current ones, deciding which predictions should be updated. The authors present an efficient probabilistic approach that answers both questions. Empirical evaluation shows the efficacy of their proposed approach over baselines.


**Main Review:**

The paper introduces a very important problem and, in general, is very clearly written and is very easy to follow. The proposed method is simple and intuitive. Limitation of the current work is also presented succinctly.

Comments:
- Three desiderata are postulated in a very clear manner. Figure 1 illustrates the approach very clearly.
- The proposed problem and solution are contrasted nicely with the related work.
- Empirical results are strong in experiments across multiple common benchmark datasets.

Limitations:
- As mentioned by the authors, the assumption of conditionally independent classifiers is strong as usually the improvements are built on prior architectures.


**Time Spent Reviewing:**

5

---

> ### Author Response · Authors · 2021-08-10
> **Author response to y17T**
>
> We thank the reviewer for their time spent reviewing our work and are happy to see they liked the submission.
>
> **“the assumption of conditionally independent classifiers is strong as usually the improvements are built on prior architectures”**
>
> In summary, Assumption 1 was made to ensure tractable inference which is important as we only have limited labelled data to estimate the confusion matrix elements $\pi^t(i,k)$.
> It was important to us to explicitly state this assumption. This model choice is an approximation but also note that any parametric distribution would still be a model approximation. The main reason for this choice is a) tractable posterior inference and b) tractable parameter estimation.
> Note that even with this model assumption the number of parameters to estimate for the full confusion matrix case $\pi^t(i,k)$ is already quickly becoming infeasible: num_classes^2 parameters, which is 1M for ImageNet. A less restrictive model choice that additionally incorporates the dependence on M previous classifiers would result in estimating the elements of a tensor of dimensionality M+2, which yields an exponential number of parameters (num_classes^(M+2)). To tame this we would need to either regularize or put parametric assumption on the distribution in Eq.(1) which would again be an approximate model.
> We believe that this choice is a good compromise between model fit and tractability.

---

### Official Review · Reviewer_j1CS · 2021-07-19

**Rating:** 8
**Confidence:** 4

**Summary:**

This paper proposes an approach for updating predictions on an unlabeled dataset as new, pretrained classifiers become available. Calling it the "Prediction Update problem", the authors seek to 1) make updates which improve accuracy, 2) avoid updates which introduce new errors, and 3) update predictions on data points using a limited budget (relevant in the massive dataset setting). The authors propose a Bayesian approach in which a posterior belief over true labels for each sample are updated as new classifier predictions become available for that sample. The authors assume (label-conditional) independence of the classifiers, allowing the likelihood to be computed easily using normalized confusion matrices of the classifiers. Samples are chosen for prediction on the basis of maximum posterior entropy, and different prediction update strategies are considered. These include naive baselines (e.g., newest classifier's prediction and majority vote) as well as MAP estimates and cost-aware strategies (which tradeoff large belief increases against the possibility of incorrectly flipping a prediction). The authors extensively discuss limitations and opportunities for future improvements. Detailed experiments using imaging benchmark datasets examine the properties of different strategies on standard (i.e., train and test data from same dataset/distribution) and transfer learning (i.e., train and test data from different datasets) tasks.

**Limitations And Societal Impact:**

Yes.

**Main Review:**

This paper considers an interesting and, to my knowledge, new problem. Existing work on backwards compatibility in ML has focused on training new models whose predictions are consistent with the old model (as discussed by the authors in Section 5). Instead, this work considers the perspective of an owner of, e.g., a proprietary unlabeled dataset for which they need labels to provide downstream services. It is increasingly common for large, state-of-the-art pretrained models (e.g., winners of benchmark challenges) to be publicly released. This work is aimed towards those who want to capitalize on the increasing availability of high quality pretrained models to perform labeling on their own data. It is a very interesting use case and I'm glad to see the authors begin a thorough investigation here.

The proposed approach is a very straightforward, but elegant, Bayesian approach for tracking and updating posterior belief about the true labels in a backwards compatible way. The proposed framework lends itself to a number of natural extensions (e.g., synthesizing probabilistic predictions instead of classification predictions, new selection strategies, adaptive budgets, etc.). I thought Section 4 on extensions and limitations was very well thought out and it addressed a number of the minor questions/concerns that I thought of while reading the paper. Finally, I thought the experiments were well executed and were a convincing demonstration of the proposed approach. The experiment studying the order of the classifiers is particularly interesting and important---I think the authors should draw more attention to it since I doubt new classifiers will truly be strictly better than older ones.

I have a few suggestions for improvement:
- When first reading the introduction I was a little confused by the setting, and had a hard time imagining possible use cases. The authors first discuss the problem in general and only then introduce the photo tagging example on Line 79. I think this should be reversed: first provide the concrete motivating example and then explain the general problem/principles that will drive the method and the rest of the paper.
- Is there any way to test Assumption 1 or measure the magnitude of violations? In Section 4 the authors mention this assumption is a limitation, but could you provide suggestions or possible avenues for addressing it, measuring it, or performing sensitivity analysis?
- Is my understanding correct that no adjustments to the method were made for the ImageNet --> ObjectNet experiment? That is, the updates were performed assuming no distribution shift? I think it would be beneficial for the authors to expand their discussion in Section 4 of how distribution shift affects the prediction update problem (and also discuss beyond just the covariate shift case), since this is a problem that is going to come up a lot in practice.


UPDATE:
Thanks to the authors for their response. I remain very positive about this work and will be keeping my score the same. I think slightly more detailed discussion of Assumption 1 and distribution shifts in the main paper will help round it out nicely.

**Time Spent Reviewing:**

2.5

---

> ### Author Response · Authors · 2021-08-10
> **Author response to j1CS**
>
> We thank the reviewer for the positive feedback and the additional suggestions for improvements.
>
> **Placement of the motivating example.**
>
> In the past, we have received mixed feedback on whether to start with the formal definition or the motivating example, with a slight majority favouring the former. We will reconsider this choice and try to further improve the Introduction.
>
> **On the assumption of conditionally independent classifiers and its violation in practice.**
>
> In the case of two given classifiers our assumption implies that $\forall n, \forall (t,t'): I(\hat{Y}^t_n; \hat{Y}^{t'}_n|Y_n)=0$. To quantify deviations of Assumption 1, one could therefore measure this conditional mutual information, but note that this in itself is challenging on finite data and would again inherit model assumptions through the choice of the estimator. Since pairwise independence does not imply mutual independence, even having these quantities might not be sufficient for a long classifier sequence. An alternative approach could thus be to measure the Maximum Mean Discrepancy (MMD) between the LHS and RHS of Assumption 1, which can be viewed as a conditional version of the d-dimensional Hilbert Schmidt Independence Criterion (dHSIC), see [1].
>
> Importantly, all of the above approaches aim to quantify and subsequently incorporate correlations between classifier predictions / confusion matrices, and thus rely on having large amounts of labelled data to properly estimate them which is almost never the case in practice. Since our goal was to devise an approach that can be useful in practice, when the number of classes, observations, and models is typically very large, we believe that the model choice behind Assumption 1 is reasonable; it enables both tractable inference and parameter estimation.
>
>
> [1] Pfister, N., Bühlmann, P., Schölkopf, B., & Peters, J. (2018). Kernel‐based tests for joint independence. Journal of the Royal Statistical Society Series B, 80 (1), 5-31.
>
>
> **“Is my understanding correct that no adjustments to the method were made for the ImageNet --> ObjectNet experiment?”**
>
> Yes, this is correct. We applied the exact same method on the subset of ObjectNet that has images with the 113 classes that are shared with ImageNet, corresponding to a subset of 18,547 images. ObjectNet images exhibit more realistic variations than those in ImageNet, i.e. rotations, background and viewpoint such that there should be some underlying distribution shift, but this is nontrivial to quantify. We thus assume the covariate shift case, where the conditional label distribution is shared across source and target distributions. Since we believe this is one of the first papers on the Prediction Update Problem, we decided to focus on the most fundamental scenarios and left the extension to distribution shift for future work. We agree that it will help any reader and practitioner to expand on this with an additional paragraph contextualising covariate shift, concept shift or label shifts in section 4.

---

### Author Response · Authors · 2021-08-10
**General response to reviewers**

We thank all reviewers for their time and positive feedback and are pleased that they all recommend acceptance. Reviewers consider our problem setting “very important” (y17T), “new“ (j1CS) and ”(very) interesting“ (BRHA, j1CS) and the proposed method ”elegant” (j1CS), “simple and intuitive” (y17T). Our experiments are described as “well executed [and] a convincing demonstration” (j1CS), and the corresponding results as “tell(ing) a clear story” (BRHA) and “empirical[ly ...] strong [...] across multiple common benchmark datasets” (y17T). Reviewer BRHA concludes: “this work could be useful across industries, in particular high stakes situations and human trust in AI systems”.

We also thank the reviewers for their constructive suggestions for improvements. We will address individual points in our direct responses to the respective reviewers and will improve the submission as indicated in our replies.

---

### Decision · Program_Chairs · 2021-09-27

**Decision:**

Accept (Poster)

**Comment:**

This work introduces a Bayesian approach to the prediction update problem. In it they consider an interesting use case in which they want to label a large unlabelled dataset while taking advantage of new pertained classifiers as they become available. Overall, reviewers were positive about the paper, but had several good suggestions when it comes to improving the presentation. I encourage the authors to carefully incorporate reviewer feedback at the revision stage.